# Interventions in Graph Neural Networks Lead to New Neural Causal Models

## Abstract

Causality can be described in terms of a structural causal model (SCM) that carries information on the variables of interest and their mechanistic relations. For most processes of interest the underlying SCM will only be partially observable, thus causal inference tries leveraging the exposed. Graph neural networks (GNN) as universal approximators on structured input pose a viable candidate for causal learning, suggesting a tighter integration with SCM. To this effect we present a theoretical analysis from first principles that establishes a more general view on neural-causal models, revealing several novel connections between GNN and SCM. We establish a new model class for GNN-based causal inference that is necessary and sufficient for causal effect identification. Our empirical illustration on simulations and standard benchmarks validate our theoretical proofs.

## 1 Introduction: Causality + Machine Learning = Understanding?

If we take causality and somehow sensibly combine it with modern machine learning like deep learning, will it lead to (deep) *understanding*?[1] With this work we intend on moving one step closer to a conclusive answer.

Understanding causal interactions is central to human cognition and thereby of high value to science, engineering, business, and law (Penn & Povinelli, 2007). Developmental psychology has shown how children explore similar to the manner of scientist, all by asking "What if?" and "Why?" type of questions (Gopnik, 2012; Buchsbaum et al., 2012; Pearl & Mackenzie, 2018), while artificial intelligence research dreams of automating the scientist's manner (McCarthy, 1998; McCarthy & Hayes, 1981; Steinruecken et al., 2019). Deep learning has brought optimizable universality in approximation which refers to the fact that for any function there will exist a neural network that is close in approximation to arbitrary precision (Cybenko, 1989; Hornik, 1991). This capability has been corroborated by tremendous success in various applications (Krizhevsky et al., 2012; Mnih et al., 2013; Vaswani et al., 2017) and most recently with the advent of large-scale models like GPT-3 (Brown et al., 2020) or DALL-E 2 (Ramesh et al., 2022). Thereby, combining causality with deep learning is of critical importance for research on the verge to a human-level intelligence. Preliminary attempts on a tight integration for so-called *Neural Causal Model* (NCM; Zečević et al. (2021); Xia et al. (2021); Pawlowski et al. (2020)) exist and shows to be promising towards the dream of a system that performs causal inferences at the same scale of effectiveness as modern-day neural modules in their most impressive applications. Any of the previously listed references present an original interpretation or account of what the word 'NCM' should refer to specifically. In this work, without stating a formal definition, we simply adopt the view that any Structural Causal Model that is being parameterized by neural models (e.g. Multi-Layer Perceptron [MLP], Recurrent Neural Network, Sum-Product Networks, etc.), in some systematic manner, is a NCM. While this informal definition is rather loose as we do not define 'systematic', we argue it to be intuitive and sensible since any of the listed references would still be considered NCM as desired. For instance, it is systematic to use MLPs to model each structural equation.

While causality has been thoroughly formalized within the last decade (Pearl, 2009; Hernán & Robins, 2010; Peters et al., 2017), deep learning on the other hand saw its success in practical applications with theoretical

---

[1]Note how this phrase is taken as a visionary account by Pearl himself in several of his public appearances. It is the foundational question of whether causal models are indeed what is needed for the "next-generation of learning systems."

breakthroughs remaining in the few. Bronstein et al. (2017) pioneer the notion of geometric deep learning and an important class of neural networks that follows from the geometric viewpoint and generalize to modern architectures is the *Graph Neural Network* (GNN; Veličković et al. (2017); Gilmer et al. (2017)). Similar to other specialized neural networks, the GNN has resulted in state-of-the-art performance in specialized applications like drug discovery (Stokes et al., 2020), biological networks (Lecca, 2021), ETA prediction in google maps (Derrow-Pinion et al., 2021), roads networks (Jepsen et al., 2020) or general density estimation in the form of variational graph auto-encoder (VGAE; Kipf & Welling (2016a)). These specialities, to which we refer to as inductive biases, can leverage otherwise provably impossible inferences (Gondal et al., 2019). As the name suggests, the GNN places an inductive bias on the structure of the input i.e., the input's dimensions are related such that they form a graph structure. To link back to causality, at its core lies a *Structural Causal Model* (SCM) which is considered to be the model of reality responsible for data-generation. The SCM implies a graph structure over its modelled variables, and since GNN work on graphs, a closer inspection on the relation between the two models seems reasonable towards progressing research in the emerging field centered around neuro-causal-symbolic AI (Zečević et al., 2022).

Instead of taking inspiration from causality's principles for improving machine learning (Mitrovic et al., 2020; Magliacane et al., 2018), we instead show the reverse direction by investigating to which extend and under which conditions machine learning is capable of causal reasoning. Specifically, we show how GNN can be used to perform causal computations i.e., how causality can emerge within graph-structured neural models. The term of "causal computation" is simply to be understood as an implementation of a causal model that can be queried. To be more precise on the term causal inference: we refer to the modelling of Pearl's Causal Hierarchy (PCH). That is, we are given partial knowledge on the SCM in the form of e.g. the (partial) causal graph and/or data from the different PCH-levels. To this end, we establish a new model class, partial causal models (PCM as we will call them from here on out), for GNN-based causal inference that is necessary and sufficient for causal effect identification and propose *interventional variational graph autoencoders* (iVGAE), the first GNN based PCM. When referring to "necessary and sufficient conditions" we mean our effort of identifying and characterizing the criteria upon which we can indeed claim to have performed a valid causal inference with GNN models. Therefore, this work is concerned with foundational questions of the abstraction level of models opposed to specific effect identification in any particular data set. To introduce this notion of PCM, we first establish a novel theoretical connection between graph neural networks and structural causal models. It is important to note that since we consider parital knowledge, this work is *not* concerned with other popular forms of causal inference like for example causal discovery (inferring the graph structure to begin with). However, this does not imply that the presented results can't be used for these other types of inferences, quite to the contrary, we are hopeful that our paper lays ground work for such investigations since initial links between structure learning (not necessarily causal) and GNNs have already been uncovered (Yu et al., 2019).

Overall, we make a number of key contributions:

C1. To further complete the existing views on neural-causal models, we show to which extent but also under which conditions GNN become representable as SCM and vice versa.

C2. We discover three new neural-causal models that emerge naturally out of C1. being: (i) the first *partially* causal model based on GNN, called interventional VGAE (iVGAE), (ii) a maximally parameterized NCM which employs neural models for each causal relation pair separately, and (iii) a minimally parameterized NCM that hides away the structural equations in its implicit message-passing reminiscent of a single GNN.

C3. For the iVGAE from C2.(a) we provide theoretical results on the feasibility, expressivity, and identifiability alongside an extensive illustration in different empirical settings.

While we consider our theoritical investigation the key contribution, we also corroborate those with empirical validation. For reproduction of the latter, we make our code repository publicly available: `https://anonymous.4open.science/r/TMLR-Submission-Causality-and-Graph-Neural-Networks-5D52`.

The paper is further structured into four sections in the following way:

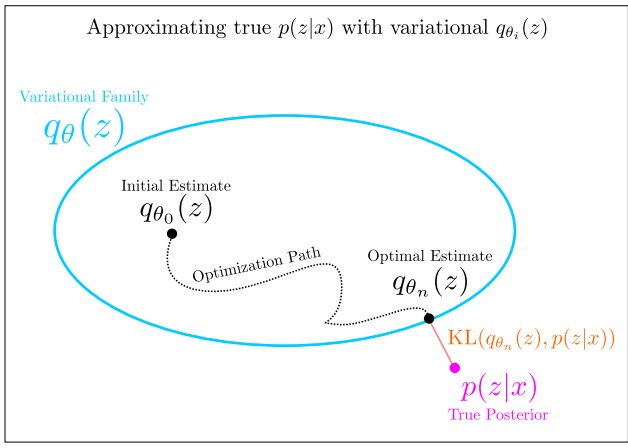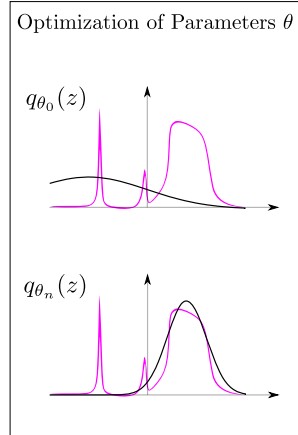

Figure 1: **Variational Inference.** A schematic overview. Left, shows the variational family set $\mathcal{Q}$ where $n$-step optimization reveals $q_{\theta_n}$ with parameters $\theta_n$ that best models the target $p$ in terms of KL-divergence. Right, illustrates the example if $Q$ were the set of Gaussian distributions trying to fit a complicated posterior $p$. Eventually, we find the best possible approximation to $p$ when using $\mathcal{Q}$. (Best viewed in color.)

**S1** Since this work makes use of concepts and works from Variational Inference, Geometric Deep Learning and Causality literature, we will start by discussing essential notions and related work first.

**S2** We explore the relations between GNN and (parameterized) SCM in a systematic manner, discovering and discussing the models from C2.

**S3** We then analyze iVGAE specifically in more detail since it poses a first partially causal model, a notion that we introduce and whose properties we try to capture formally.

**S4** Finally, we provide an empirical illustration on available data sets corroborating the main theoretical analysis. The empirical part is divided into two subsections (1) a discussion of general probabilistic density estimation and (2) of causal effect estimation.

## 2 Background and Related Work: VI, GNN, Causality

Before presenting our main theoretical findings, we briefly review the background on (i) variational methods for generative modelling, on (ii) graph neural networks as non-parametric function approximator that leverage structural information as inductive bias, and conclusively on (iii) causal inference through the process of intervention/mutilation in Pearl's theory.

**Notation.** We generally follow standard notation forms from (Goodfellow et al., 2016; Pearl, 2009). We denote indices by lower-case letters, functions by the general form $g(\cdot)$, scalars or random variables interchangeably by upper-case letters, vectors, matrices and tensors with different boldface font $\mathbf{v}, \mathbf{V}, \mathsf{V}$ respectively, and probabilities (both discrete/continuous) of a set of random variables (RV) $\mathbf{X}$ as $p(\mathbf{X})$. In slight abuse of notation parameters $\theta, \boldsymbol{\theta}$ might be dropped or used interchangeably if clear from context. PCH levels (or languages) are denoted with $\mathcal{L}_i, i \in \{1, 2, 3\}$, the potential outcome or counterfactual is denoted as $Y_x$ where $x \in \mathrm{Val}(X)$ is the instance of RV $X$ and a pure intervention (only $\mathcal{L}_2$ quantity) of any type: soft, perfect, atomic via the *do*-operator like $do(X = x)$.

**Variational Inference.** Similar to the notions of disentanglement and causality, latent variable models propose the existence of a-priori unknown, essential variables $\mathbf{Z}$ to jointly model them with the phenomenon of interest for which we have some observed data, that is, $p(\mathbf{X}, \mathbf{Z})$. Put slightly differently, the meaning of latent variables in latent variable models lies in them being some sort of "underlying, generative factors" for our data. This is the interpretation of SCMs from causality (to be discussed in the paragraph following GNNs) and similarly in disentanglement we usually try the same, that is, find factors that are separate and explanatory of our data. The Variational Inference (VI) technique makes use of *optimization*, as an alternative

to Markov Chain Monte Carlo sampling (MCMC) approaches, for overcoming the "curse of dimensionality"[2] when estimating probability distributions (Jordan et al., 1999; Blei et al., 2017). To provide clarification on how VI might achieve overcoming the cure: on an intuitive level we make intractable queries tractable (or at least approximative) by using optimization. In reality, we really just re-formulate the problem to something that is just more practical for downstream tasks. Since we are in a Bayesian context, the inference problem amounts to estimating the latent variable conditional $p(\mathbf{Z} \mid \mathbf{X})$ (posterior). This is done through finding the closest density of a pre-specified family $\mathcal{Q}$, that is:

$$q^*(\mathbf{Z}) = \arg \min_{q \in \mathcal{Q}} \mathrm{KL}(q(\mathbf{Z}) \,\|\, p(\mathbf{Z} \mid \mathbf{X})), \tag{1}$$

where the distance measure is set to be the Kullback-Leibler (KL) divergence. The model family $\mathcal{Q}$ is usually chosen to be 'neat' in that it is satisfying desirable computational properties. For instance we might choose $\mathcal{Q}$ to be the family of Gaussian Mixture Models (Marin et al., 2005). Fig.1 illustrates the idea schematically by showing visually how such an optimization might come about. It is worthwhile noting that the overall problem described in Eq.1 is *intractable* in the average setting, since inspecting Bayes Rule exposes that $p(\mathbf{Z} \mid \mathbf{X}) = \frac{p(\mathbf{X}, \mathbf{Z})}{p(\mathbf{X})}$ where the so-called evidence in the denominator is an exponential term in $\mathbf{Z}$, that is, $p(\mathbf{X}) = \int p(\mathbf{X}, \mathbf{Z}) \, d\mathbf{Z}$. Luckily, using Jensen's inequality, we can derive a tractable lower bound. Originally derived in (Jordan et al., 1999), said bound is placed on the evidence and is revealed to be

$$\log p(\mathbf{X}) - \mathrm{KL}(q(\mathbf{Z}) \,\|\, p(\mathbf{Z} \mid \mathbf{X})) = \mathbb{E}_q[\log p(\mathbf{X} \mid \mathbf{Z})] - \mathrm{KL}(q(\mathbf{Z}) \,\|\, p(\mathbf{Z})), \tag{2}$$

where the first term (of the r.h.s.) expresses likelihood (or reconstruction) of the data under the given parameters while the divergence terms counteracts such parameterization to adjust for the assumed prior distribution. Choosing $p_{\boldsymbol{\phi}}(\mathbf{X} \mid \mathbf{Z})$ and $q(\mathbf{Z}) := q_{\boldsymbol{\theta}}(\mathbf{Z} \mid \mathbf{X})$ to be parameterized as neural networks leads to the well-known family of variational auto-encoder (VAE) (Kingma & Welling, 2019). For such VAE, importance sampling (Rubinstein & Kroese, 2016) brings about sampling techniques for performing marginal inference with probabilistic models i.e., since we have the equality in the limit that

$$p(\mathbf{X}) \approx \frac{1}{n} \sum_{i=1}^n \frac{p_{\boldsymbol{\phi}}(\mathbf{X} \mid \mathbf{z}_i) p(\mathbf{z}_i)}{q_{\boldsymbol{\theta}}(\mathbf{z}_i \mid \mathbf{X})}. \tag{3}$$

Fortunately, the number of samples $n$ is usually being kept moderate through the likelihood ratio induced by $q$, therefore, offering a tractable and practical method for marginal inference. While in our empirical part we do not run into the issue of scaling it should be noted that scaling with this technique is not recommended, in that sense it can be considered less practical (for further references see (Kingma & Welling, 2019)). This consistent (for $n \to \infty$), empirical approximation of the marginal $p(\mathbf{X})$ will be used in the empirical section of this paper.

**Graph Neural Networks.** In geometric deep learning, as portrayed by (Bronstein et al., 2021), graph neural networks (GNN) constitute a fundamental class of function approximator that place an inductive bias on the structural relations of the input. A GNN layer $f(\mathbf{D}, \mathbf{A}_G)$ over some data which we consider to be vector-valued samples of our endogenous variables $\{\mathbf{d}_i\}_{i=1}^n =: \mathbf{D} \in \mathbb{R}^{d \times n}$ and some adjacency representation $\mathbf{A}_G \in \{0, 1\}^{d \times d}$ of a directed acyclic[3] graph $G$ over our variables is generally considered to be a *permutation-equivariant*[4] application of *permutation-invariant* functions $\phi(\mathbf{d}_X, \mathbf{D}_{\mathcal{N}_X^G})$ on each of the variables (features) $\mathbf{d}_i$ and their respective neighborhoods within the graph $\mathcal{N}_i^G$. The most general form of a GNN layer (also known as the message-passing GNN) is specified by

$$\mathbf{h}_i = \phi\left(\mathbf{d}_i, \bigoplus_{j \in \mathcal{N}_i^G} \psi(\mathbf{d}_i, \mathbf{d}_j)\right), \tag{4}$$

---

[2]Uniformly covering a unit hypercube of $n$ dimensions with $k$ samples scales exponentially, that is, $O(k^n)$.

[3]Note that for computation purposes it is common practice to allow each node a self-loop to keep its own representation during computation. Other than that, the graphs used are acyclic. Still, as Bongers et al. (2021) have shown, causal models might not necessarily need to be considered acyclic as has been long-standing tradition in the causality literature since having acyclic graphs allows for a great deal of mathematical proofs to become easier and classes beyond so-called 'simple' SCM need to be studied more.

[4]That is, for some permutation matrix $\mathbf{P} \in \{0, 1\}^{d \times d}$, it holds that $f(\mathbf{PD}, \mathbf{PA}_G\mathbf{P}^\mathsf{T}) = \mathbf{P}f(\mathbf{D}, \mathbf{A}_G)$.

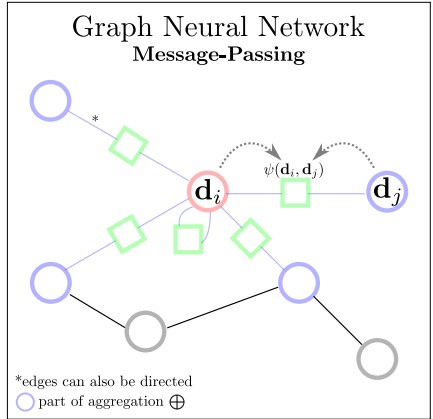 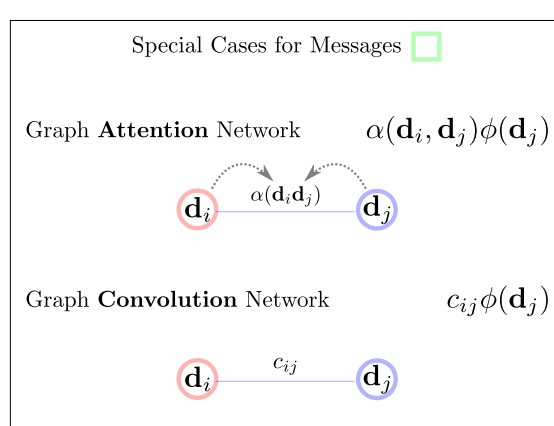

Figure 2: **Graph Neural Networks.** A schematic overview. Left, shows the computation of a single vector-valued node (red) as an aggregation of its neighborhood's (blue) and own information (cycle) through messages (green). Right, illustrates the special cases: GAT (top) with implicit attention-based weights for node computation, and the simplest case GCN (bottom) with constant weights. Note that 'simple' does not directly equate to any performance measure, all three flavors pose trade-offs. (Best viewed in color.)

where $\mathbf{h}_i$ represents the updated information of node $i$ aggregated ($\bigoplus$) over its neighborhood in the form of messages $\psi$. The functions $\phi, \psi$ are usually learnable. To give an example, $\phi$ could be a ReLU activated linear combination ($\phi(\mathbf{x}) = \sigma(\mathbf{Wx} + \mathbf{b})$ with learnable parameters $\mathbf{W} \in \mathbb{R}^{n \times |\mathbf{x}|}, \mathbf{b} \in \mathbb{R}^n$) function whereas $\psi$ could be a plain linear combination of feature vectors and $\bigoplus_{i \in A} i := \sum_{i \in A} i$ simply summation (but it could also be mean, maximum or a more sophisticated algorithmic operation schemes). This general 'flavour' of GNN presented in Eq.4 is being referred to as message-passing (MP-GNN; Gilmer et al. (2017)) and constitutes the most general class of GNN that supersets both convolutional (Kipf & Welling, 2016a) and attentional (Veličković et al., 2017) flavours of GNN. Note how the message type ($\psi$) is being shared for all the variables ($\forall i$), whereas the message itself will depend on the values of the variable $i$ and its neighbors. This aspect of the computation rule might be considered the key defining property of a GNN. We illustrate the computation rule in Eq.4 and the hierarchy of the three flavors alongside their pioneering models Graph Attention Networks (GAT) and Graph Convolutional Networks (GCN) schematically in Fig.2. In the context of representation learning on graphs, GCN (Fig.2 right bottom) were previously used within a VAE pipeline as means of parameterization to the latent variable posterior $p(\mathbf{Z} \mid \mathbf{X})$ (Kipf & Welling, 2016b). The authors combined VI with GNN, forming the V**G**AE, to perform inference on graph structured data effectively. A similar techique was recently deployed in learning dynamical systems from purely observational data (Kipf et al., 2018). Note that the three flavors of GNN, that is MP-GNN, GAT and GNN, pose different dis-/advantages for different applications. E.g. while MP-GNN allows for the highest representational capacity, as it is subsuming the other two, GCN might still outperform MP-GNN in an application due to various other factors such as for instance the cost of optimization. While in our theoretical discussion we consider any flavor type of GNN, for our empirical part we will mostly resort to simple GCN because of their lower representational complexity which in turn shrinks the room of interpretation possibility when discussing evidence.

**Causal Inference.** Following the Pearlian notion of Causality (Pearl, 2009), an SCM is defined as a 4-tuple $\mathcal{M} := \langle \mathbf{U}, \mathbf{V}, \mathcal{F}, P(\mathbf{U}) \rangle$ where the so-called structural equations

$$v_i = f_i(\mathrm{pa}_i, u_i) \in \mathcal{F} \tag{5}$$

assign values (denoted by lowercase letters) to the respective endogenous or system variables $V_i \in \mathbf{V}$ based on the values of their parents $\mathrm{Pa}_i \subseteq \mathbf{V} \setminus V_i$ and the values of their respective exogenous[5] variables $\mathbf{U}_i \subseteq \mathbf{U}$, and

---

[5]These variables are sometimes also referred to as 'unmodelled', 'nature' or 'noise' variables. In essence, they describe everything that we cannot (or don't want to) model explicitly. Note that the probability measure is defined on these exogenous terms, which implies a distribution over the endogenous variables. The mechanistic, structural equations are deterministic.

$P(\mathbf{U})$ denotes the probability function defined over $\mathbf{U}$. For readers more familiar with the ML literature: endogenous ('inside' system variables) are typically what we call features, whereas exogenous ('outside' system variables) could be considered as anything that is latent. However, the latter distinction is not as simple, since we cannot equate exogenous variables to, say, hidden units in a neural net since they have well-defined semantics. On an intuitive level, however, this is how the distinction in variable type can be understood from a ML perspective. An intervention on an SCM occurs when (multiple) structural equations are being replaced through new non-parametric functions thus effectively creating an alternate SCM. Before introducing interventions formally, we note that we will refer to our base distribution, the "observational distribution w.r.t. SCM $\mathcal{M}$," as $p^{\mathcal{M}}$. Interventions are referred to as *perfect* if the parental relation is cut, as *soft* if not but the relation itself is changed, and even *atomic* when additionally to being perfect the intervened values are being kept to some constant. Hard intervention and atomic intervention are terms used synonymously and they refer to the act of putting the value of an endogenous variable to some scalar. Mathematically denoted $do(X = a)$ with $a \in \mathbb{R}$. It is important to realize that interventions are of fundamentally *local* nature, and the structural equations (variables and their causes) dictate this locality. This further suggests that mechanisms remain *invariant* to changes in other mechanisms. An important consequence of said autonomic principles is the *truncated factorization*

$$p^{\mathcal{M}_{do(\mathbf{w})}}(\mathbf{v}) = p^{\mathcal{M}}(\mathbf{v} \mid do(\mathbf{w})) = \prod\nolimits_{V_i \notin \mathbf{W} = \mathbf{w}} p^{\mathcal{M}}(v_i \mid \mathrm{pa}_i) \tag{6}$$

derived by Pearl (2009), which suggests that an intervention $do(\mathbf{w})$ introduces an independence of a set of intervened nodes $\mathbf{W}$ to its causal parents. We use $p^{\mathcal{M}}$ and $p$ interchangeably if clear from context. For completion we mention more interesting properties of any SCM: (a) they induce a causal graph $G$ typically but not necessarily as directed acyclic graph (DAG), (b) they induce an observational also called associational distribution denoted $p^{\mathcal{M}}$, and (c) they can generate infinitely many *interventional* and *counterfactual* distributions. Note that, opposed to the Markovian SCM discussed in for instance (Peters et al., 2017), the definition of $\mathcal{M}$ in the presented setting is semi-Markovian (or non-Markovian) thus allowing for shared $U$ between the different $V_i$. Put in different words, non-Markovian SCM simply refers to an SCM where endogenous variables can share their exogenous variables. I.e., we have hidden confounders. Markovian SCM then means that there are no hidden confounders. Such a $U$ is also called *hidden confounder* since it is a common cause of at least two $V_i, V_j (i \neq j)$. Opposite to that, a 'common' confounder would be a common cause from within $\mathbf{V}$. In this sense, the SCM extends the (Causal) Bayesian Network (CBN; Pearl (2011)) since it allows beyond interventions for both counterfactuals and reasoning about hidden confounders (Bongers et al., 2021). Because of that, SCM are sometimes referred to as *Functional* Bayesian Networks. The SCM's applicability to machine learning has been shown in marketing (Hair Jr & Sarstedt, 2021), healthcare (Bica et al., 2020) and education (Hoiles & Schaar, 2016) to name just a select few applications. As suggested by the Causal Hierarchy Theorem (CHT; Bareinboim et al. (2020)), the properties of an SCM form the Pearl Causal Hierarchy (PCH) consisting of different levels of distributions being $\mathcal{L}_1$ *associational*, $\mathcal{L}_2$ *interventional* and $\mathcal{L}_3$ *counterfactual*. The PCH suggests that causal quantities $(\mathcal{L}_i, i \in \{2, 3\})$ are in fact richer in information than statistical quantities $(\mathcal{L}_1)$, and the there exists a necessity of causal information (e.g. structural knowledge, essentially 'outside' model knowledge) for inference based on lower rungs e.g. $\mathcal{L}_1 \not\Rightarrow \mathcal{L}_2$ and therefore to reason about $\mathcal{L}_2$ or to *identify* such causal quantities we need more than only observational data from $\mathcal{L}_1$. These levels or languages[6] differ in that $\mathcal{L}_1$ is common statistics $p(\mathbf{A})$, $\mathcal{L}_2$ are expressed through interventions $p(\mathbf{A_b})$ (this notation of a potential outcome denotes "the value of $\mathbf{A}$ had $\mathbf{B}$ been $\mathbf{b}$" and is used subsequently to represent different worlds for the counterfactual case, the relation to the regular *do*-operator is described at the paragraph's end) and $\mathcal{L}_3$ are conjunctions of the former $p(\mathbf{A_b}, \ldots, \mathbf{C_b})$. Finally, a last note regarding simulating SCM to acquire actual data: to query for samples of a given SCM, the structural equations are being simulated sequentially following the underlying causal structure starting from independent, exogenous variables $U_i$ and then moving along the causal hierarchy of endogenous variables $V_i$. To conclude this paragraph on causality, consider the formal definition of valuations for the highest layer $(\mathcal{L}_3)$ since it subsumes the other two layers as previously pointed out:

$$p(\mathbf{a_b}, \ldots, \mathbf{c_d}) = \sum_{\mathcal{U}} p(\mathbf{u}) \quad \text{where} \quad \mathcal{U} = \{\mathbf{u} \mid \mathbf{A_b}(\mathbf{u}) = \mathbf{a}, \ldots, \mathbf{C_d}(\mathbf{u}) = \mathbf{c}\}, \tag{7}$$

---

[6]Language as in logic, see Def.8 "Symbolic Languges" in (Bareinboim et al., 2020).

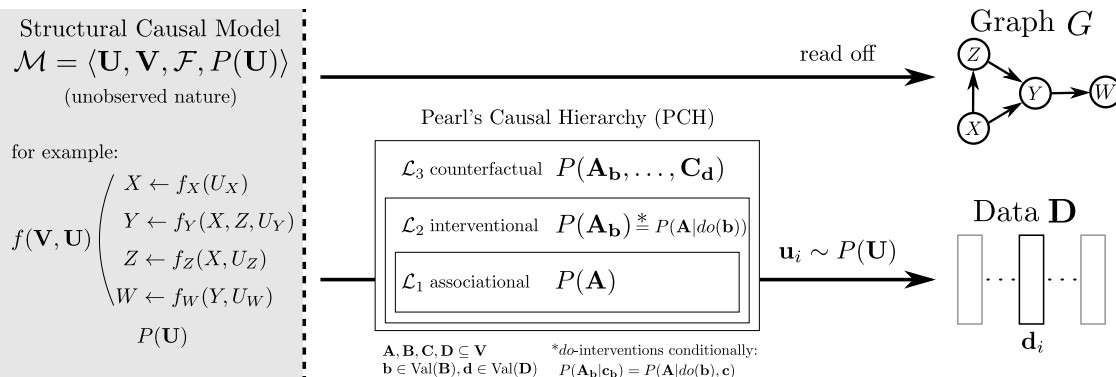

Figure 3: **SCM and PCH.** A schematic overview. Left, shows the generally hidden SCM underlying the phenomenon of interest. Middle, the implied PCH. Right, the implied graph and data distributions upon evaluating 'nature' terms $U_i$ and subsequently structural equations. (Best viewed in color.)

for instantiations $\mathbf{a}, \mathbf{b}, \mathbf{c}, \mathbf{d}$ of the node sets $\mathbf{A}, \mathbf{B}, \mathbf{C}, \mathbf{D} \subseteq \mathbf{V}$ and they represent different 'worlds.' E.g. for $\mathcal{L}_1$ we might only consider $\mathbf{A} = \mathbf{A}_\emptyset$ and whereas for $\mathcal{L}_2$ a single alternate world $\mathbf{A}_\mathbf{b}$. Note that the $do(\mathbf{x})$ notation and the counterfactual subscript notation $(\cdot)_\mathbf{x}$ only coincide as $p(\mathbf{a}_\mathbf{b} \mid \mathbf{c}_\mathbf{b}) = p(\mathbf{a} \mid do(\mathbf{b}), \mathbf{c})$ which is related to whether the condition $\mathbf{c}$ is pre- or post-treatment (Pearl, 2009).

Causal assumptions for parts of this work: since this work is the first to study the relation of GNNs and SCMs mathematically, we choose to generally consider the classical causal inference setting of Markovian SCMs. This assumption implies that the structure is acyclic, there is no latent confounding and so the observational distribution is unique. We will also mostly focus on the classical notion of hard intervention where values are set to constants. Both assumptions are common practice in causality literature when providing a theoretical analysis on bounds, identifiability and similar. Our main reason to pledge to those is similarly simplicity in presentation and proofs. However, we do foresee that we need not be strict with several discussions we raise in this work e.g., when discussing interventions with iVGAE we can safely ignore the second assumption for a lot of cases.

**A Remark on Similarities to Bayesian Deep Learning.** In the past there have also been efforts on the foundational front regarding the integration of Bayesian approaches with Deep Learning, specifically Bayesian Networks. Since BNs are the ancestor to SCMs in that SCMs extend them (a) to have causal semantics i.e., the edges in the graph implied by an SCM, opposed to a BN, are causal and (b) we can compute counterfactuals and talk about hidden confounders, we can therefore see our discussion as an integration of causality with deep learning, being Causal Deep Learning.

**Further Related Work.** While this work is neither concerned with causal *discovery* nor with pure *identification*, both of these areas are central to causality. A recent original work on the former proposed a meta-objective based on a notion of *speed of adaptation* which will allow the true causal structure to adapt 'faster' in response to the sparse change (Bengio et al., 2019). For the latter, a pragmatic, high dimensional setting was investigated in (Jesson et al., 2021). Only recently, important work has also been done on the intersection of both, where an extension of the 'standard' notion of the Pearlian causal inference framework to multiple data sets with each having different contexts was presented (Mooij et al., 2020). A special case of this setting focussed on investigating the cases where the source and target data distributions differ (Magliacane et al., 2018). This has paved way for causality to be applied to an extended set of machine learning models. One such approach has been suggested by (Zaffalon et al., 2020) where equivalences to credal networks were established. Furthermore, the usage of GNNs within causal domains have only recently been studied and have stayed either at the application level (Wein et al., 2021; Li et al., 2020) or have been used for generating causal explanations for the underlying GNNs (Lin et al., 2021; Bajaj et al., 2021). In this work, we take an inspection of how causality *per se* 'naturally' arises in GNN and their relation to SCM, NCM and aspects of *partial* causality.

## 3   Allowing Interventions in GNN and Minimal/Maximal NCMs

To expand further on the boundaries of the integration between causality and machine learning, we perform a theoretical investigation on the relation between graph neural networks (GNN) and structural causal models (SCM), thereby transitively also their common parameterized variant: neural causal models (NCM). Note how when we use NCM without any further specifications, we generally refer to *any* type of NCM (models that can be considered NCM), which will include the two special types to be discovered in this section. While all the established results on causal identification have proven that intervention (sometimes also referred to as manipulation) is not necessary for performing causal inference, the concept of intervention still lies at the core of causality as suggested by the long-standing motto of Peter Holland and Don Rubin *"No causation without manipulation"* (Holland, 1986). The centrality of interventions is why we choose to consider them as a starting point of our paper's discussions. It is important to note that choosing to start with or even keep the discussion restricted to interventions does not imply that we might only connect GNN to Causal Bayesian Networks (CBN; see Pearl (2011)), which model causality but only up to interventions (ignoring counterfactuals and confounders). Historically, SCM extended CBN by the notion of *counterfactuals* and *hidden confounders* (Bongers et al., 2021) and many times in the literature SCM may also be identified as *Functional* BN (FBN) to denote the lineage with regular (Causal) Bayesian Networks. Most important for the distinction is the fact that the exogenous variables can be modelled explicitly. To return to our previous point, therefore, having interventions in GNN would lead to a connection to CBN but also leave open a connection to full SCM–which we will also consider in this work–but for simplicity, we start with interventions. To this effect, we first define a process of intervention within the GNN computation layer that will subsequently reveal sensible properties of the process akin to those of intervention on SCM.

**Definition 1** (Intervention on GNN Computation)**.** *An intervention $\mathbf{x}$ on the corresponding set of variables $\mathbf{X} \subseteq \mathbf{V}$ within a GNN layer $f(\mathbf{D}, \mathbf{A}_G)$, denoted by $f(\mathbf{D}, \mathbf{A}_G \,|\, do(\mathbf{X} = \mathbf{x}))$, is defined as a modified layer of computation,*

$$\mathbf{h}_i = \phi\left( \mathbf{d}_i, \bigoplus_{j \in \mathcal{M}_i^G} \psi(\mathbf{d}_i, \mathbf{d}_j) \right), \tag{8}$$

*where the intervened local neighborhood is given by*

$$\mathcal{M}_i^G := \mathcal{N}_i^G \setminus \mathrm{pa}_i \quad if \quad i \in \mathbf{X} \quad else \quad \mathcal{N}_i^G \tag{9}$$

*where $\mathcal{N}^G$ denotes the regular graph neighborhood. Such GNN-layers are said to be interventional.*

Note that we focus on directed edges as common in causality, still the general GNN formalism would actually be capable of bi-directedness. However, allowing for those would conflict with the meaning of a *direct cause* as indicated by a direct edge in a causal graph, thereby leaving them out of the definition. The above definition is akin to a hard intervention and takes into account the self-loop (since $\mathbf{d}_i$ for node $i$ is always part of the computation). An intervention on a GNN, just like in an SCM, is local in nature i.e., the new neighborhood of a given node is a subset of the original neighborhood at any time, $\mathcal{M} \subseteq \mathcal{N}$. The notion of intervention belongs to the causal layers of the PCH i.e., layers 2 (interventional) and 3 (counterfactual). Fig.4 presents an intuitive illustration of how the semantic changes within the SCM upon intervention translate to a change in computation for the GNN. A causal relationship is always based on directionality in that one variable is called the 'cause,' whereas the other is known as the 'effect,' which is to be contrasted with general computational models like the GNN, where we usually look at the neighborhood of a node thus ignoring the directionality and even computing using values that in the causal sense would be considered coming from a child node. In our notion of (hard) intervention, the key property that we use is that parental relations are being severed.

The motivational origin of this work lies in the tighter integration of causality with present machine learning methodologies like neural networks. Ultimately, we envision a fully-differentiable system that combines the benefits of both worlds. We can see Def.1 as a first step towards this goal, since it allows us now to view interventions in GNN as a special case of *parameterizing a CBN* with neural approximators. Remembering C2 of our contributions, we suggested that we had discovered three new neural-causal model classes, namely (i) the first *partially* causal model based on GNN, (ii) a *maximally* parameterized NCM, and (iii) a *minimally* parameterized NCM. We will start off by taking a conservative view on SCM exploring (i-iii) in *reverse* order, where we will begin with (iii) and then gradually drop assumptions/requirements to conclude with (i) where

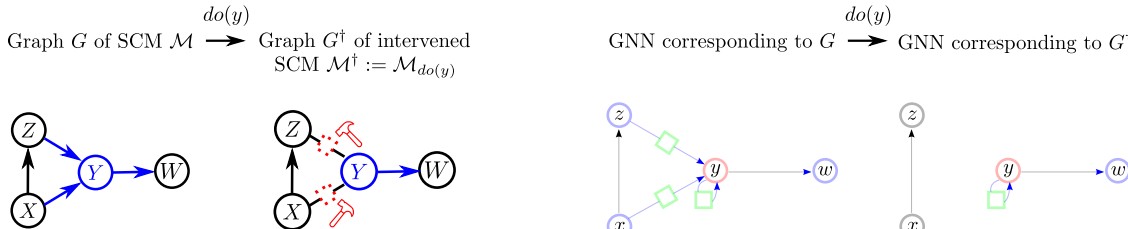

Figure 4: **Graph Neural Networks and Interventions akin to SCM.** A schematic overview. Left, a *do*-intervention on an SCM. Right, the same for a GNN according to Def.1. Semantically, we cut the causal dependence on the parents (for atomic interventions $y$), which in turn cuts the number of necessary computations in the GNN. (Best viewed in color.)

we spend the bulk of our discussion and our empirical investigation. We find (iii) by asking the question: with interventions now at our disposal, can we also talk about counterfactuals (quantities on $\mathcal{L}_3$ of the PCH) and thereby a relation between GNN and fully fledged SCM? Indeed, it turns out we can consider a direct conversion between GNN and SCM. Such a conversion between GNN and SCM will necessarily have to cope with transforming a shared, global function $\psi$ into the collection of all local partial mechanisms $f_{ij}$ of any structural equation[7]. Thereby, one can anticipate that training such model becomes tremendously difficult due to this *globality* constraint of the GNN. More formally, we can still state the following model conversion.

**Theorem 1** (C2(iii), Minimal NCM, Existence of SCMs in the Set of All GNNs)**.** *Consider a message-passing GNN node computation $\mathbf{h}_i : \mathcal{F}_h \mapsto \mathcal{F}'_h$ as in Eq.4. For any SCM $\mathcal{M} := \langle \mathbf{U}, \mathbf{V}, \mathcal{F}, P(\mathbf{U}) \rangle$ there always exists a choice of feature spaces $\mathcal{F}_h, \mathcal{F}'_h$ and shared functions $\phi, \psi$, such that for all structural equations $f_i \in \mathcal{F}$ it holds that $\mathbf{h}_i = f_i$.*

*Proof.* It is sufficient to provide a general construction scheme on SCMs. Therefore, let $\mathcal{M} := \langle \mathbf{U}, \mathbf{V}, \mathcal{F}, P(\mathbf{U}) \rangle$ be any SCM. Further, let $f_i(\mathrm{pa}_i, U_i) \in \mathcal{F}$ be the structural equation of endogenous variable $V_i \in \mathbf{V}$. We can now consider a decomposition of the structural equation of the form $f_i(\mathrm{pa}_i, U_i) = f_{ii}(U_i, \mathcal{A}_i) + \sum_{j \in \mathrm{pa}_i} f_{ij}(V_j)$ where the summation is over all linearly decomposable causal sub-effects $f_{ij}(V_j)$ (what we referred to as 'partial' previously) whereas the first term depends on a dynamic, potentially empty argument list $\mathcal{A}_i \in 2^{|\mathrm{pa}_i|}$ that captures all aspects of the structural equation $f_i$ that are not decomposable in linear terms. For instance, if $f_i$ is linear in its arguments, then $\mathcal{A}_i = \emptyset$, but if the structural equation $f_i$ might be something like $f_i(V_i, U_i) = V_i \cdot U_i$, then $\mathcal{A}_i = \{V_i\}$. This decomposition is similar in spirit to what can be found in for instance (Kuo et al., 2010). It is only important to note that whether $\mathcal{A}_i$ compromises the empty set or the maximal set of parents of $V_i$, that is $\mathrm{pa}_i$, is not critical for establishing the proof but rather for talking about the similarity of the given SCM and GNN. I.e., if $\mathcal{A}_i = \mathrm{pa}_i$, then we might consider the SCM and GNN to be maximally similar in that each causal sub-effect $f_{ij}$ maps to a message computation in the GNN layer $\mathbf{h}_i$. Having said that, now, we are equipped to map between GNN and SCM. We choose the following mapping for the respective GNN computation components:

$$\mathcal{F}_h, \mathcal{F}'_h := \mathbf{V} \cup \mathbf{U} \tag{10}$$

$$\phi(\mathbf{d}_i, \bigoplus_{j \in \mathcal{N}_i^G} \psi(\mathbf{d}_i, \mathbf{d}_j)) := f_{ii}(U_i, \mathcal{A}_i) + \sum_{j \in \mathrm{pa}_i} \psi(\mathbf{d}_i, \mathbf{d}_j) \tag{11}$$

$$\psi(\mathbf{d}_i, \mathbf{d}_j) := f_{ij}(\mathbf{d}_j). \tag{12}$$

Note how having $\psi$ not depend on $\mathbf{d}_i$ in $f_{ij}$ is simply an artefact of notational conventions, since in causality we typically use subscripts to denote the specific function and for causal sub-effects we then use both variables $(i, j)$ that compose the message for the GNN message-passing function $\psi$. Further, note how $\mathcal{N}_i^G = \mathrm{pa}_i$ holds because of the decomposition according to the causal graph $G$ implied by $\mathcal{M}$. Finally, it holds that

$$\mathbf{h}_i = \phi\bigg(\mathbf{d}_i, \bigoplus_{j \in \mathcal{N}_i^G} \psi(\mathbf{d}_i, \mathbf{d}_j)\bigg) = f_{ii}(U_i, \mathcal{A}_i) + \sum_{j \in \mathrm{pa}_i} f_{ij}(V_j) = f_i. \tag{13}$$

---

[7]Here 'partial' refers to the 'structure' of any single structural equation. In a linear SCM, we can simply imagine this to be the coefficients or causal effects $f_{ij} := a_j \cdot x_j$, where $f_i := \sum_{j \in \mathrm{pa}_i} a_j \cdot x_j$. In the case that some effects are shared e.g. when there is a multiplicative component like $f_{i\{jk\}} := a_{jk} \cdot x_j \cdot x_k$, the separation still works but is naturally less decomposed.

Note how the notation on $f_i$ is purposefully overloaded to make clear the correspondence between the new latent vector and the corresponding structural equation result, $f_i$ is to be understood as $f_i(u_i, \mathrm{pa}_i)$.    □

The common ground between SCM and GNN lies within their graph structures, which is typically implied for the SCM since we learn it and assumed for the GNN since it acts as a prior or inductive bias. This common ground in the graph is in fact the key in leveraging the reparameterization from GNN to SCM as our Thm.1 suggests. We can simply use the mappings dictated by the graph to establish the relation by decomposing the structural equation into said causal sub-effects. Let us illustrate on an example how this might look like.

**Example 1** (Two SCMs and their Sub-Effects). *First let us consider the case where we get a complete or maximal decomposition of a structural equation into sub-effects. This occurs for instance when the structural equations of the SCM are linear in its arguemnts. For instance, when*

$$\mathcal{M}_1 = (\{f_X(Z, U_X) := Z + U_X, \ f_Z(U_Z) := U_Z\}, \quad P(U_X, U_Z)), \tag{14}$$

*then we have $\mathcal{A}_X = \emptyset$ and simply choose to separate $f_X$ into two functions $f(U_X) = U_X, f(Z) = Z$. Now, as a second case, let us consider the other end of the extreme being the case where we cannot easily decompose our SCM's structural equations i.e., our argument list from Thm.1 is the maximal set, the set of $V_i$'s parents $\mathcal{A}_i = \mathrm{pa}_i$. For instance,*

$$\mathcal{M}_2 = (\{f_X(Z, U_X) := Z \wedge U_X, \ f_Z(U_Z) := U_Z\}, \quad P(U_X, U_Z)), \tag{15}$$

*where $\wedge$ is a logical AND operator ($X, Z$ are binary). Then we have*

$$f_X = \begin{pmatrix} \overset{U_X=0}{0} & \overset{U_X=1}{1} \\ 0 & 0 \end{pmatrix} \begin{matrix} Z=1 \\ Z=0 \end{matrix} \iff \begin{matrix} f_{XZ}(Z) + f_{U_X}(U_X, Z) \\ [Z] + [U_X - (Z \vee U_X)]. \end{matrix} \tag{16}$$

*The decomposition in (16) simply takes the identity of the parent ($Z$) while considering the negated logical OR for the remainder term $f_{U_X}$. Its sum then results in the original logical AND function $f_X$ of $\mathcal{M}$. While the decomposition for this specific example does not seem to reveal any sensible advantage as opposed to the linear case, it still is captured by Thm.1. With this we have seen that the actual decomposition (and therefore $\mathcal{A}_i$) will lie somewhere on this spectrum from 'empty' to 'containing all parents' for any given variable. The wording 'actual' here refers to the specific $\mathcal{A}$ that one will encounter when considering the conversion for some SCM-GNN tuple. Thus it reveals a natural tendency towards the style of computation of 'messages' in the GNN. Since in a GNN we typically consider the complete neighborhood of a variable, a linear SCM would be the most natural choice for such a computation.*    ■

Before moving to the insights that we can extract from the above example, we want to once more highlight that $\mathcal{A}$ also handles non-linear SCM, that is, we pose no restrictions on our original SCM definition. Nonetheless, the linear SCM can be considered a clear special case for when $\mathcal{A}$ contains all the variables since we have coefficients equaling the causal sub-effects. Up until now we have established two new insights. One, how to define interventions on the computation within GNN akin to SCM which compares them directly to at least the class of causal models such as CBN. Two, how to reparameterize SCM to do computation akin to GNN, therefore, making GNN directly comparable to even the class of complete causal models such as SCM. With Thm.1 we extend the previously existing big picture in the literature on neural-causal models since we can now understand GNN as another 'species' of neural SCMs and more specifically our theorem provides a constructive approach that illustrates the necessary conditions required for taking that perspective. Actually, we can even be more specific than that since these 'GNN-SCM' are actually *minimally* parameterized neural SCM. We can understand this by considering the intuition behind the proof. Essentially, what happens in the proof is that we construct a sort of 'look-up table' of messages for the GNN in advance. The messages in this table are the sub-effect decompositions of each of the SCM's structural equations and then $\psi$ in the GNN computation is simply able to reference for each variable pair the corresponding sub-effect.[8] By doing

---

[8]Since structural equations are not required to be decomposable, we need the technically around the remainder term or argument list $\mathcal{A}$ but that is not important for the discussion of minimality in parameterization.

so, we ultimately have an SCM that hides away its structural equations implicitly within the message passing of a single GNN. It is important here to understand that an SCM (and typical formulations of NCM) are *sets of models* decomposing according to a graph, whereas a GNN is just a *single model* depicting a graph. In that sense our construction in Thm.1 is truly the *minimal* NCM (C2 iii) since we replace multiple graph-structured neural nets with one single message-passing GNN. Interestingly, we can now view our newly discovered neural-causal model also as an insight on GNNs in that there exist *causal models within the set of all GNNs.* A more radical view would postulate that *all* GNNs are therefore SCMs (since the Theorem proves the existence of a SCM for any GNN), just that the SCMs that are being modelled by our GNN are most of the time not the SCMs that we seek for our application.[9] Before turning to the next thought, let us re-iterate once more on this in different terms. Indeed our Theorem shows a correspondence between the sets of SCM and the sets of GNN, however, there is a gap between just having a computational model and the 'right' model. SCMs are usually confused to be the "underlying reality," although they are simply a formalism for talking about causality. Any given SCM could potentially be the "underlying reality" for some phenomenon but this potential is in no way a necessity. To grasp once more the foundational insight provided by our Theorem: under consideration of $\mathcal{A}$, we get to know how we can map between computations (or 'messages') in a GNN and causal effects in an SCM. This $\mathcal{A}$ will be very different for different structural equations of (different) SCMs. Furthermore, it raises the point of differentiation between "purely computational" and causal models. Turning now to a different aspect of our theorem, it does not give away any information on *optimization.* Put differently, Thm.1 does not talk about whether the reparameterization is *feasible* in the practical sense when being deployed as a machine learning model. It follows naturally that $\psi$ is a *shared* function amongst all nodes of the graph while an SCM considers a specific mechanism *for each* of the nodes in the graph. This subtle, yet crucial distinction is what makes optimization difficult for a GNN that mimics an SCM in the way that Thm.1 proposes. In a nutshell, the messages $\psi(i,j)$ need to model each of the causal sub-effects $f_{ij}$ within a structural equation such that the messages themselves become a descriptor of the causal relation for $V_i \leftarrow V_j, \{V_i, V_j\} \subseteq \mathbf{V}$. The question arises whether there is a smarter way of reparameterization of SCM in terms of GNN. Can we for instance drop the *sharedness* property of GNN? Unfortunately, we cannot if we intend on preserving the GNN since the shared message-computing function is essential to what defines a GNN. Nonetheless the answer to this research question poses an interesting insight on the class of all neural SCMs. We observe that dropping sharedness by allowing each of the $\psi(i,j)$ to be computed by a separate function $\psi_{ij}$ reveals a new NCM model that opposed to C2(iii) is *maximally* parameterized (C2(ii)) since now we can use local messages parameterized by MLP[10] that allow for computation *per-variable.* This stands in contrast to 'classical' NCM that use a MLP each for modelling each of the SCM's structural equations. They are also *the maximal* NCM model class since it poses the most fine-grained parameterization using MLPs that can be chosen for parameterizing SCM and therefore lies at the other end of the spectrum of parameterization as the previous GNN-SCM (Thm.1) which only used a single GNN for modelling all of the structural equations. Accordingly, we state the following:

**Corollary 1** (C2(ii), Maximal NCM)**.** *Assume the setting from Thm.1. Further, we allow for the violation of sharedness of $\psi$ by arbitrary parameterization with MLP. The resulting computation is an NCM special case with a computation per-edge.* ∎

The proof can be found in the appendix. In the special (or extreme) case of the original SCM being linear, the maximal NCM portrayed in Cor.1 is then maximally more fine-grained than the the definition of classical NCM since the former will model each of the causal sub-effects $f_{ij}$ with separate MLPs, whereas the original NCM will choose only the structural equations to be modelled by MLPs i.e., the NCM is an aggregated or consolidated view of the maximal NCM. Naturally, this makes the latter a computationally more expensive model since the architecture will scale with both with the number of equations *but also their decomposability.* In a sense, NCM will scale with the number of endogenous variables $\mathcal{O}(|\mathbf{V}|)$ whereas maximal NCM scale with the number of edges $\mathcal{O}(|\mathcal{E}|)$ of the SCM's implied graph $G_\mathcal{M} = (\mathbf{V}, \mathcal{E})$. We believe that maximal NCM can justify their overhead in computation through improved interpretability since each causal sub-effect can be investigated separately to attribute on the cause level instead of the mechanism level. Note that this

---

[9]This is partly a philosophical discussion on the topic whether causality is a concept independent of *truth.* That is, there are a lot of SCM that are 'gibberish' in that they don't model our physical reality faithfully (i.e., they are not good 'models'), however, they are still causal since they follow the definition of an SCM and endogenous variables are computed through structural equations.

[10]In this work we write MLP = multi-layer perceptron = (feed-forward) neural net = NN.

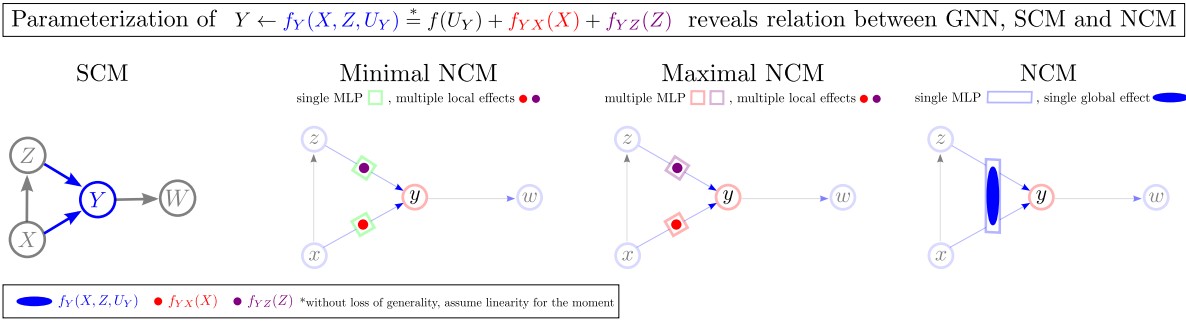

Figure 5: **Reparameterization Reveals Relation Between GNN, SCM and NCM.** A schematic overview of the results established in Thm.1 and Cor.1. The design choice can be summarized as (a) modelling on mechanism or cause level, and (b) modelling with a single, shared or multiple independent function approximators. (Best viewed in color.)

discussion around NCM is one of parameterization, that is, actual *computational* models for SCM. Therefore, the discussion is not concerned with learning the actual graph iteratively using an NCM or GNNs, which was explored by for instance (Ke et al., 2019) and (Lin et al., 2021) respectively.

To conclude this section, we point to Fig.5 which illustrates schematically how the parameterization of a structural equation reveals different connections between GNN, SCM and NCM. In a sense, the design choices involve whether to model on mechanism (classical NCM as in previous literature) or on cause level (Minimal and Maximal NCM) and further whether to model with multiple (Maximal NCM) or only a single approximator (Minimal and classical NCM).

**Contextualizing the Different NCM Variants.** The minimal and maximal NCM pose original contributions that define extensions to the original NCM as formally discussed in (Xia et al., 2021). The extensions come in the form of extreme ends on the dimension of *number of neural modules employed*, further justifying the use of the terms 'minimal' (for the least number of neural modules, that is, one) and 'maximal' (for the highest number of neural modules, that is, the number of edges in the implied causal graph).

## 4  iVGAE: A First Partially Causal Model Based on GNNs

When we say *partially* causal, then we refer to models that lie on the spectrum that is spun by non-causal models such as linear regression, CNNs, Transformers etc. and causal models such as fully-fledged SCM but that are belonging neither to the former nor the latter class. For readers more familiar with ML literature it might come as a surprise as labelling certain models as 'causal' or 'non-causal' since in their perspective the *type of inference* (here causal) is usually considered independent of the model employed. This is especially true if we consider for example that linear models can be used *for* causal inference. Or in the same way, that an SCM can consist of linear equations. However, the semantics are inherently different. An SCM actually is a causal model since the equations denote what it means to be 'causal,' that is, if they are not, then we don't have an SCM. This is not the case for linear models for instance, which is why we refer to them as non-causal. Furthermore, like depicted in Fig.6, 'full' causal means that we fully capture the hierarchy i.e., you have an SCM. That is, "fully causal" is synonymous to being capable of generating all 3 types of distributions. In other words, these models are partially causal because they induce some causal inference capabilities but it is clear that they are not SCM equivalent. Looking again at the newly uncovered Minimal NCM in Thm.1 and the Maximal NCM in Cor.1 (both of which are depicted in Fig.5), we observe that they are parameterized variants of an SCM and thus *fully* causal models. Furthermore, both of these models are difficult to handle computationally, which makes them more restrictive. For said two reasons, being theoretically unappealing and computationally prohibitive, we decide to focus in this section on what can be achieved when following only our initial definition of interventions within GNN (see Def.1)—which will reveal the *first partially causal model based on GNN*. Since such partially causal models, have not been investigated theoretically in the literature, we provide a first attempt. To the best of our knowledge, the first (albeit non-GNN) PCM discussed in the literature was the interventional Sum-product Network from

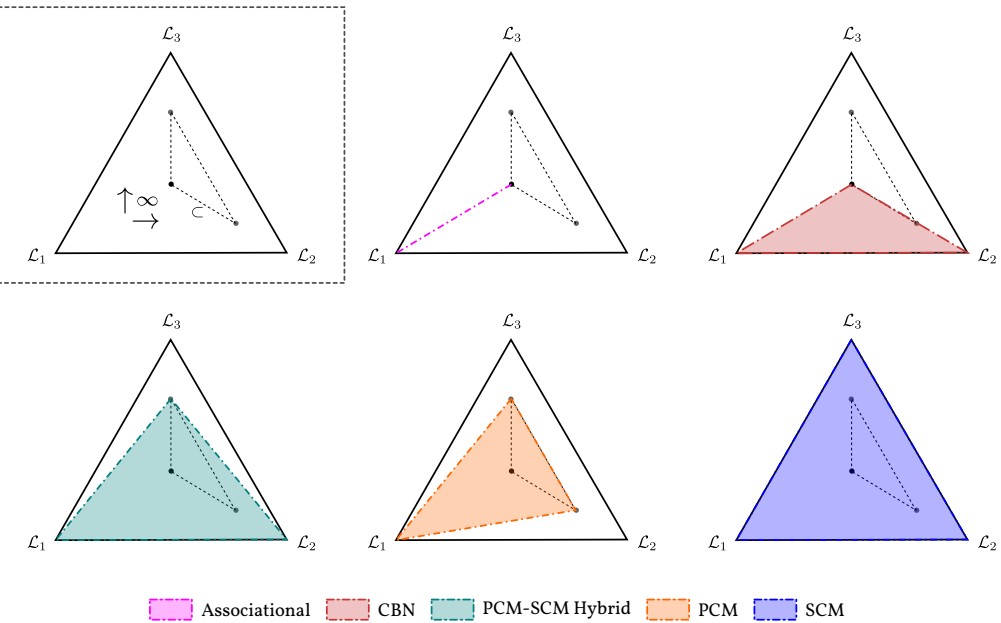

Figure 6: **Classifying Causal Models based on PCH Expressivity.** Legend (top left): $\mathcal{L}_i$ denote the PCH levels, mid-point denotes $\mathcal{L}(\text{model}) \subset \mathcal{L}_i$, whereas outer-point denotes $\mathcal{L}(\text{model}) = \mathcal{L}_i$. Partial Causal Models are those models that span an area that is neither minimal (Associational) nor maximal (SCM). Note how the classical CBN falls into the category of PCM. (Best viewed in color.)

(Zečević et al., 2021), even if the authors did neither define PCM nor classify their model as such. To this end, we will consider a construction of a GNN based on partial knowledge on the SCM like the SCM's implied graphical structure. We define a GNN construction based on an SCM as follows:

**Definition 2.** *Let $G_{\mathcal{M}}$ be the graph induced by SCM $\mathcal{M}$. A GNN layer $f(\mathbf{D}, \mathbf{A}_G)$ for which $G = G_{\mathcal{M}}$ holds is said to correspond to SCM $\mathcal{M}$. If $\mathcal{M}$ is clear from context, we drop the subscript.*

Remember that $G$ (from $\mathbf{A}_G$) is simply the inductive bias, in the form of a graph (adjacency), to the GNN, whereas $G_{\mathcal{M}}$ (in alternate literature also $G(\mathcal{M})$) is the graph implied by some SCM $\mathcal{M}$. This definition allows for providing prior knowledge on the causal structure of the problem as a basis for the GNN to be deployed. This definition is purposefully inline with common practices in GNN literature but it further comes with the subtle notion on the fact that *any* GNN actually models *some particular* SCM. A simple consequence of constructing the previous Definitions 1 and 2 is the following:

**Proposition 1.** *Let $\mathcal{M}$ be an SCM with graph $G$ and let $f$ be a GNN layer corresponding to $\mathcal{M}$. An intervention $do(\mathbf{X}), \mathbf{X} \subseteq V$, on both $\mathcal{M}$ and $f$ produces the same intervened graph.* ■

Like before the proof is in the appendix. The above proposition essentially acts as a sanity check to us choosing sensible definitions prior since now an intervention within a GNN layer (Def.1) is *dual* to the notion of intervention within an SCM i.e., like observing within the intervened graph is equivalent to performing interventions (probabilistically $p^{\mathcal{M}}(\mathbf{v} \mid do(\mathbf{W})) = p^{\mathcal{M}(do(\mathbf{W}))}(\mathbf{v})$), computing a regular GNN layer on the intervened graph is equivalent to performing the intervention on the original GNN layer, that is, $f(\mathbf{D}, \mathbf{A}_G \mid do(\mathbf{X})) = f(\mathbf{D}, \mathbf{A}_{G_{\mathbf{x}}})$ where $G_{\mathbf{x}}$ is the graph upon intervention $do(\mathbf{X})$. For readers familiar with optimization literature, 'dual' here should be simply understood as the two perspectives of GNN and SCM type of interventions, which through Prop.1 are shown to be equivalent (which further justifies that the definition of GNN intervention was chosen sensibly.) Now, before defining our PCM based on a GNN that we constructed from an SCM, we first need to consider how to relate PCM to SCM in more detail. That is, how can we compare the causal aspects of a PCM with the causal aspects of an SCM? To answer this question, we take inspiration from prior works that defined notions of *consistency* for matching the different distributions that are being emitted by the different layers $\mathcal{L}_i$ of the PCH (see Fig.3 and discussions in Sec.2)

by any given model. To this end, we now define what it means for an SCM to be consistent with a model that is *not an SCM*.

**Definition 3** (PCM). *Let $\mathcal{M}$ be an SCM and $\mathcal{L}_i(\mathcal{M})$ the set of all distributions implied by $\mathcal{M}$ on level $i$ of the PCH (see Sec.2, e.g. $\mathcal{L}_2$ allows for all distributions with $do(\mathbf{X})$ where $\mathbf{X} \subseteq \mathbf{V}$). We say that a model $\mathcal{V}$ is partially consistent with $\mathcal{M}$ if either (i) or (ii) holds:*

*(i) $\mathcal{L}_i(\mathcal{V})$ is defined only for $i \in \{1,2\}$ and $\mathcal{L}_i(\mathcal{V}) \subseteq \mathcal{L}_i(\mathcal{M})$ with $|\mathcal{L}_i(\mathcal{V})| \in [1..\infty]$*

*(ii) $\mathcal{L}_i(\mathcal{V})$ is defined for $i \in \{1,2,3\}$ and $\mathcal{L}_i(\mathcal{V}) \subset \mathcal{L}_i(\mathcal{M})$ with $|\mathcal{L}_i(\mathcal{V})| \in [1..\infty)$.*

*Further, $\mathcal{V}$ is called a partially causal model.*

Note how the definition requires $\mathcal{V}$ to be a model capable of implying/emitting distributions, so it can be any generative, probabilistic model such as for instance the popular class of variational auto-encoders. Furthermore, said model should fulfill any of the two conditions: Def.3(ii) $|\mathcal{L}_i(\mathcal{V})| \in [1..\infty)$ is bounded and not infinite i.e., there is no procedure for generating either infinitely many interventional or counterfactual distributions, or Def.3(i) $\mathcal{V}$ is not capable of producing counterfactuals in the first place. These conditions are *necessary conditions* for these models to be considered partial as otherwise it would be either an standard SCM or a model capable of reasoning across the complete causal hierarchy which is equivalent to an SCM in terms of causal reasoning.[11] A consistency w.r.t. some SCM $\mathcal{M}$ is actually enforced through the operator $\mathcal{L}_i(\mathcal{M})$ which gives the set of all distributions for level $i$ of that SCM and the PCM then only requires a subset of that specific set. Without this condition, we'd only have a PCM and not a *SCM-corresponding PCM*, which is the defining property we formalize. To make this new conceptual idea of partial causality and PCMs more clear, we provide a schematic illustration in Fig.6. In the following, we simply talk about consistency but mean partial consistency as in Def.3. As we have seen, a consistent model is therefore by definition a causal model capable of emitting a subset of the $\mathcal{L}_i$-distributions of the PCH. We are now finally ready to define our first PCM based on GNN. As the underlying skeleton, we'll use the *variational graph auto-encoder* (VGAE; see Kipf & Welling (2016b)) since they define a standard, generative probabilistic model with GNN being the underlying workhorse.

**Definition 4** (iVGAE). *Let $\mathcal{V}$ be a VGAE with encoder $q$ and decoder $p$ being GNNs. If $(q, p)$ are set to be interventional GNN layers (Def.1) modelling the latent variables and endogenous variables (data) respectively, then $\mathcal{V}$ is also called interventional VGAE.*

In Fig.7 we schematically illustrate the difference of a PCM (specfically, the just-defined iVGAE) against the traditional SCM. Both allow for causal computation (generative modelling of interventional distributions) but that is where the similarities end, since the PCM will resort to data and a gating signal provided by the intervention on the underlying causal graph, whereas the SCM will simply sample the exogenous variables and then evaluate in topological order each and every structural equation. An immediate key observation is that the *SCM is a lot more complex* in terms of model description.

That is, an SCM needs to model each structural equation and so the model complexity (here the amount of neural models if we consider classical NCM) scales with the number of endogenous variables $\mathcal{O}(|\mathbf{U}|)$ while the iVGAE will always consist of only a single encoder-decoder pair $\mathcal{O}(1)$ independent of how many endogenous variables we might want to consider. This poses a significant advantage for causal graphs on, say, social networks which have $n$ lie in the multi-million range. What we observe is therefore a *tradeoff between model expressivity and model complexity*. See Tab.1. The iVGAE's constant complexity has an advantage over the

| Model | Expr. | Complex. |
|-------|-------|----------|
| iVGAE | $\mathcal{L}_{1,2}$ | $\mathcal{O}(1)$ |
| NCM | $\mathcal{L}_{1,2,3}$ | $\mathcal{O}(|\mathbf{U}|)$ |

Table 1: **Trade-off for Different Causal Models.**

SCM's linear complexity, yet falls short when it comes to modelling capabilities on the PCH—the iVGAE specifically has not been designed to model counterfactuals, so it fulfills condition (i) from Def.4 being a

---

[11] In other words, if our model called $\mathcal{M}_1$ can generate $\mathcal{L}_2$- and $\mathcal{L}_3$-distributions as well as match a given SCM $\mathcal{M}_2$ in the limit $(\mathcal{L}_i(\mathcal{M}_1) = \mathcal{L}_i(\mathcal{M}_2)$ where $|\mathcal{L}_i(\mathcal{M}_2)| = 1$ for $i = 1$ and $\infty$ for $i \in \{2,3\}$), then that model $(\mathcal{M}_1)$ must be an SCM as well–which makes Def.3 a sensible definition for what we mean by *partially causal*. Note that for $i > 1$, the number of possible interventions is even *uncountably* infinite, e.g. take any intervention of the form $do(X = x)$ where $\text{Val}(X) := \mathbb{R}$, which immediately implies that $|\mathcal{L}_i| = 2^{\aleph_0}$.

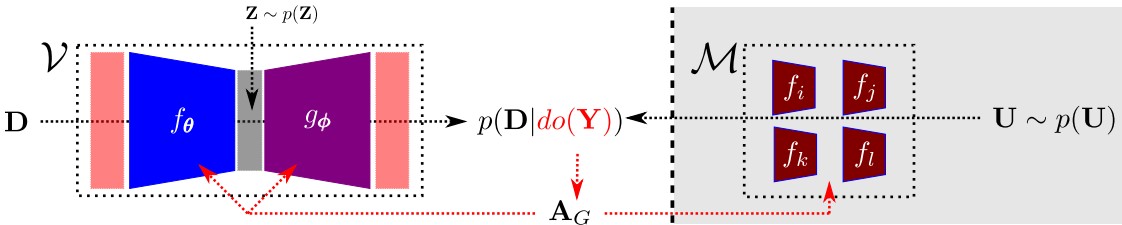

Figure 7: **Causal Inference in PCM vs SCM.** A schematic overview of the inference process within the partially causal model $\mathcal{V}$ (iVGAE from Def.4 is depicted) alongside the analogue process within a classical SCM $\mathcal{M}$. While $\mathcal{M}$ can be directly queried for the causal effect $p(\mathbf{D}|\,do(\mathbf{Y}))$ through evaluations of the exogenous terms $\mathbf{U}$, the PCM $\mathcal{V}$ makes use of corresponding mixed data $\mathbf{D}$. (Best viewed in color.)

PCM and not SCM.[12] In the following, we investigate this restricted expressivity of PCM. We first state a reassuring result on the correspondence of PCM and SCM.

**Theorem 2.** *For any SCM $\mathcal{M}$ there exists an iVGAE $\mathcal{V}$ for which $\mathcal{V}$ is $\mathcal{L}_2$ consistent w.r.t. $\mathcal{M}$.* ∎

The proof is provided in the appendix. This theorem suggests that we there will always be a (potentially infinite) number of PCM that will correspond to any SCM of interest since consistency is unbounded and we are only restricted by the available data (and knowing whether the data decomposes into interventions). It is important to note that whether we are able to actually *find* such corresponding or relevant PCM is not the topic of discussion in Thm.2. As a simple corollary, we might state that for any GNN that corresponds to an SCM (such as the Minimal NCM from Def.2) it will always allow for an iVGAE that corresponds to that very SCM that implied the graph for the GNN construction.

Just now we established iVGAE as a first instance of a GNN-based PCM capable of causal inferences, yet not being an SCM. Another important corollary is related to the *causal hierarchy theorem* (CHT; see Bareinboim et al. (2020)), which reassures that causal inference "will make sense" in that we cannot easily do inter-level jumps on the PCH. The CHT also holds for PCM (and therefore the iVGAE). We state:

**Corollary 2** (Partial-CHT for PCM)**.** *Consider the sets of all SCM and PCM, $\Omega, \Upsilon$, respectively. If for all $\mathcal{V} \in \Upsilon$ it holds that $\exists q \subset \mathbb{N}.\,[\mathcal{L}_1(\mathcal{V}) = \mathcal{L}_1(\mathcal{M}) \implies \mathcal{L}_2^q(\mathcal{V}) = \mathcal{L}_2^q(\mathcal{M})]$ with $\mathcal{M} \in \Omega$ and $q$ choosing a subset of $\mathcal{L}_2$ distributions, then we say that layer 2 collapses relative to $\mathcal{M}$. By extension of the classical NCM result on the CHT, it is clear that on the Lebesgue measure over SCMs the subset in which layer 2 of iVGAE collapses to layer 1 has measure zero.*[13] ∎

Note that the proof for the CHT has not been made publically available by the authors, however, an alternate proof from a topological perspective was recently given by a group of researchers that included a subset of the authors of the original CHT paper in (Ibeling & Icard, 2021) which supports the belief in the truth of the CHT (as also suggested in the measure theoretic sense of the original theorem which is the argument we use for Cor.2). As discussed in the literature, the CHT does not impose a negative result by claiming impossibility on lower to higher layer inferences, however, it suggests that even sufficient expressivity of a model does not allow for the model to overcome the boundaries of the layers unless causal information (e.g. in the form of structural constraints such as knowledge on the causal graph) is available. The PCHT in Cor.2 simply reassures that using PCMs does not suddenly change the familiar setting of causal inference with SCM/NCM—causal inference therefore still "makes sense" in that the different queries we can ask across the different $\mathcal{L}_i$ are indeed qualitatively different from each other. The slogan "causal inference still makes sense" is a phrase encountered often times in causality when the causal hierarchy theorem is being discussed, since this foundational result can be considered as the justification for the term 'causality' and its definition

---

[12]As we noted in the beginning, while we focus on interventions, the iVGAE could also be adapted to further condition on exogenous terms making it capable of modelling $\mathcal{L}_3$ as well since interventions are the key ingredient. However, as this work's focus is not on generative modelling specifically, we leave an extended discussion here for future work.

[13]The branch of mathematics called *measure theory* defines this notion. Probability theory is a special case of measure theory, which tries to generalize and formalize geometrical measures such as length, area or volume. To give an example of "measure zero," the rational numbers $\mathbb{Q}$ are measure zero relative to the real numbers $\mathbb{R}$ since a single point set $\{x\}, x \in \mathbb{Q}$ is measure zero and the set formed through the union over such *countably infinite* sets is still measure zero. In a sense it conveys the idea that the cases in which there could be a counterexample are essentially negligible.

by Pearl to begin with. The CHT suggests that inter-level inferences *without assumptions* are impossible, thus giving different semantics to the different levels, since if they were infer-able from each other, then that would necessarily mean that they are equivalent.

While we have not discussed identification within PCM yet, we realize that an extended discussion blows the boundaries of this initial work on GNN, SCM, NCM and finally PCM. Therefore, we leave this for future work. However, one important thing to note is that *identification and estimation actually coincide for iVGAE* (and we believe more generally for PCM, again, this needs to be investigated thoroughly in a separate work) since the model is being fed experimental data and has no inherent causal conceptions apart from allowing a naturally defined intervention on the computation of the output (see interventional GNN layers from before). Therefore, identification and estimation in the following might be viewed interchangeably but to reduce confusion amongst classical causal inference readers we stick to the term of estimation, which generally refers to using data to acquire an actual estimate for the quantity of interest. The estimation is performed using a modified version of the variational objective in Eq.2 to respect the causal quantities, $\mathbb{E}_q[\log p(\mathbf{V}|\mathbf{Z}, do(\mathbf{W}))] - \mathrm{KL}(q(\mathbf{Z}|do(\mathbf{W}))\|p(\mathbf{Z}))$, where $\mathbf{W} \subset \mathbf{V}$ are intervened variables and $\mathbf{Z}$ denotes the latent variables. After optimizing the iVGAE model with this causal ELBO-variant, we can consider any quantity of interest dependent on the modelled levels. One interesting choice for such a query $Q$ is the average treatment effect (ATE; defined as $\mathrm{ATE}(X, Y) := \mathbb{E}[Y \,|\, do(X = 1)] - \mathbb{E}[Y \,|\, do(X = 0)])$, where the binary[14] variable $X$ is being referred to as treatment.

## 5 Empirical Analysis for iVGAE

In this final technical section, we will assist our theoretical results with an extensive set of empirical illustrations on the causal modelling capabilities of our PCM instance iVGAE. Our code is publically available for scientific reproduction at: `https://anonymous.4open.science/r/TMLR-Submission-Causality-and-Graph-Neural-Networks-5D52`. In the following, we will (as for most of our theoretical analysis) discuss iVGAE and not Minimal/Maximal NCMs which we leave for future work. Since our specification of the iVGAE model does not compute counterfactuals, the following analysis only considers causal quantities of interventional form. Since gold-standard SCM data sets are unfortunately not readily available in the existing literature, we will be using data sets that come with Causal Bayesian Networks (CBN) which are equivalent to SCM in our setting since we only consider queries up to the interventional level.[15]

**Remark on the Purpose of this Empirical Analysis.** Naturally, the following analysis does not serve a contribution to deep learning but is concerned with validating the theoretical insights generated previously on *foundational questions* regarding the integration of causality with (geometric) deep learning. Therefore, this section is concerned with (a) answering empirical questions on the conceptual level and (b) sanity-checking correct model specifications as a proof-of-concept.

### 5.1 Systematic Investigation on Density Estimation

**TL;DR.** Fig.8 reveals that our PCM, iVGAE, is capable of appropriately modelling causal queries. While Thm.2 predicted the mere existence of such a consistent iVGAE, our simple experiment shows that (at least in these low-dimensional settings) optimization can also identify said models.

We perform multiple experiments to answer various interesting questions. The following list enumerates all the key *questions* to be highlighted and discussed in this section:

(a) What aspects of an interventional change through *do*-intervention does the method capture?

(b) How does variance in ELBO (Eq.2) during variational optimization affect the method?

(c) When and how does the method fail to capture interventional distributions?

(d) At what degree does the performance of the method vary for different training durations?

---

[14]Without loss of generality we can extend the ATE to be categorical/continuous.

[15]SCM extend CBN in terms of counterfactuals and hidden confounding, which are the two cases that are not of interest in the following discussion. As a note that makes connection of SCM to CBN clear, an alternate name for SCM is also *Functional Bayesian Network* (FBN).

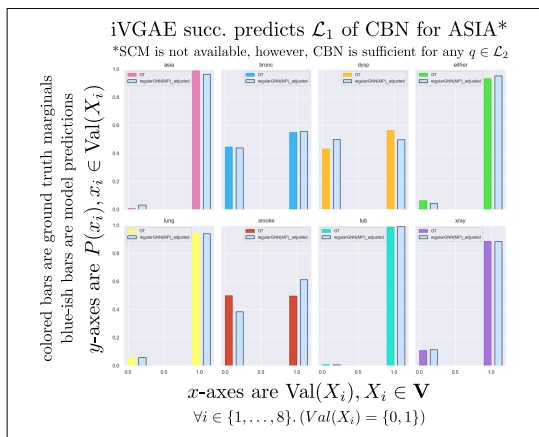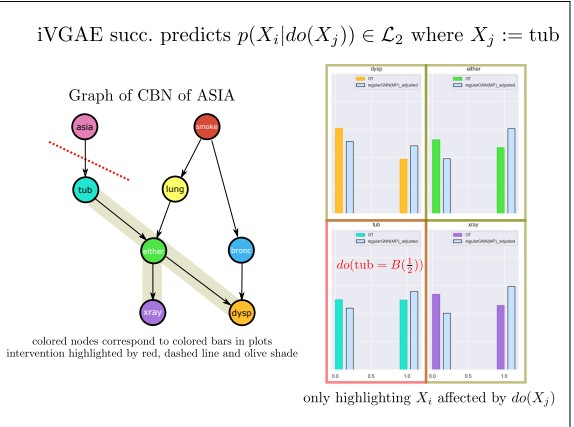

Figure 8: **Causal Density Estimation.** Illustration of the results on semi-synthetic data from the CBN `ASIA` by Lauritzen & Spiegelhalter (1988). The iVGAE adequately models both the observational and the queried interventional distributions. The intervention uses a fair coin flip (Bernoulli $\mathcal{B}(\frac{1}{2})$) to randomize the values of node `tub`. (Best viewed in color.)

    (e) How does the method scale w.r.t. interventions when capacity is kept constant?

    (f) How important is parameter tuning?

For all the subsequent experiments we considered the same architecture. That is, an iVGAE (Def.4) model consisting of two interventional GNNs (Def.1) for the encoder and decoder respectively where each GNN consists of 2 Sum-Pool Layers as introduced by (Kipf & Welling, 2016a). The decoder has $2B^2$ parameters, whereas the encoder has $3B^2$ parameters, where $B$ is the batch size, to allow for modelling the variance of the latent distribution. We consider data sets of size 10,000 per intervention. The interventions are collected by modifying the data generating process of the data sets. For simplicity, we mostly consider uniform interventions, however, without loss of generality non-perfect and soft interventions could have also been considered. Optimization is done with RMSProp (Hinton et al., 2012) and the learning rate is set to 0.001 throughout. We perform a mean-field variational approximation using a Gaussian latent distribution and a Bernoulli distribution on the output. All data sets we have considered are binary but extensions to categorical or continuous domains follow naturally.

In the following, we focus on ASIA introduced in (Lauritzen & Spiegelhalter, 1988), and Earthquake/Cancer covered within (Korb & Nicholson, 2010) respectively. Acquiring a gold standard for the causal graph, let alone the actual SCM, is difficult in practice—in the case of the data sets we consider, like ASIA, we are given the CBN which constitutes the causal graph alongside parameterizations. CBN are capable of providing interventional distributions making them a necessary and sufficient tool for our subsequent empirical analysis in which we are only concerned with observational and interventional distributions. We employ a training, validation and test set and use the validation set to optimize performance subsequently evaluated on the test set. We use a 80/10/10 split. We use 50 samples per importance sampling procedure to account for reproducibility in the estimated probabilities. Training is performed in 6,000 base steps where each step considers batches of size $B$ that are being scaled multiplicatively with the number of interventional distributions to be learned. The adjacency provided to the GNNs is a directed acyclic graph (DAG) summed together with the identity matrix to allow for self-reference during the computation. The densities are acquired using an importance sampling approach for the iVGAE. All subsequent experiments are being performed on a MacBook Pro (13-inch, 2020) laptop running a 2,3 GHz Quad-Core Intel Core i7 CPU with a 16 GB 3733 MHz LPDDR4X RAM on time scales ranging from a few minutes up to approximately an hour with increasing size of the experiments. In the following, we will have a figure each to act as reference for the subsequent subsections's elaborations on the questions (a)-(f), so Fig.9 for question (a), Fig.10 for question (b) and so on. Numerical statistics are provided in Tab.2. For reproducibility and when reporting aggregated values (e.g. mean or median) we consider 10 random seeds.

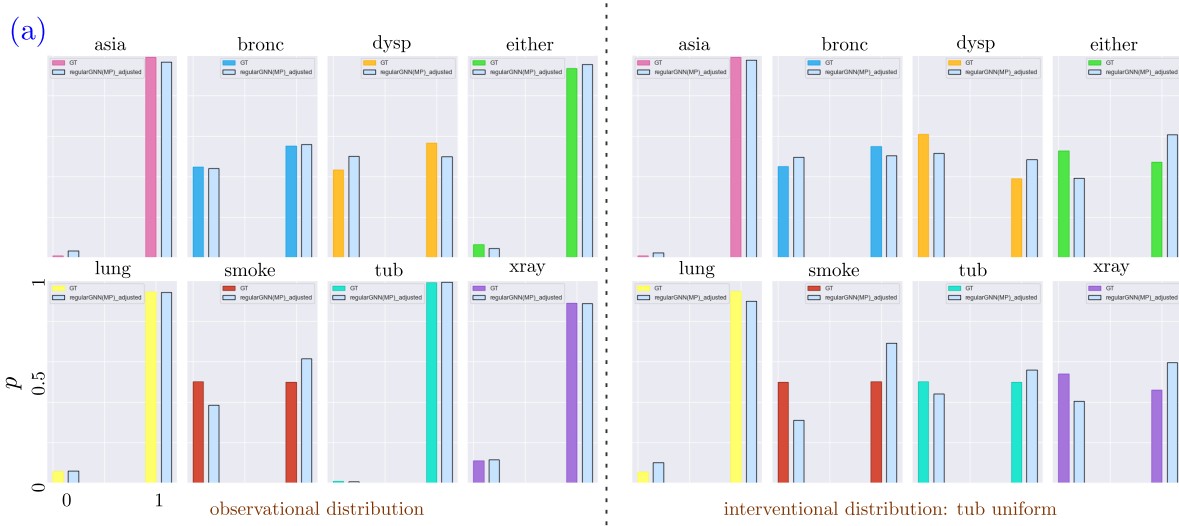

Figure 9: **Systematic Investigation: Question (a).** What aspects of an interventional change through *do*-intervention does the method capture? (Best viewed in color.)

**Q-(a) What aspects of an interventional change through *do*-intervention does the method capture?** Consider Fig.9 in the following. It shows an iVGAE model trained on the observational ($\mathcal{L}_1$) and one interventional ($\mathcal{L}_2$, intervention $do(\text{tub} = \mathcal{B}(\frac{1}{2}))$) distributions, where the former is shown on the left and the latter on the right. We can observe that both the change within the intervention location (tub) but also in the subsequent change propagation along the causal sequence (either, xray, dysp) are being captured. In fact, they are not only being detected but also adequately modelled for this specific instance. If the optimization is successful in fitting the available data with the available model capacity, then this is the general observation we make across all the other settings we have evaluated i.e., the model can pick-up on the interventional change without restrictions.

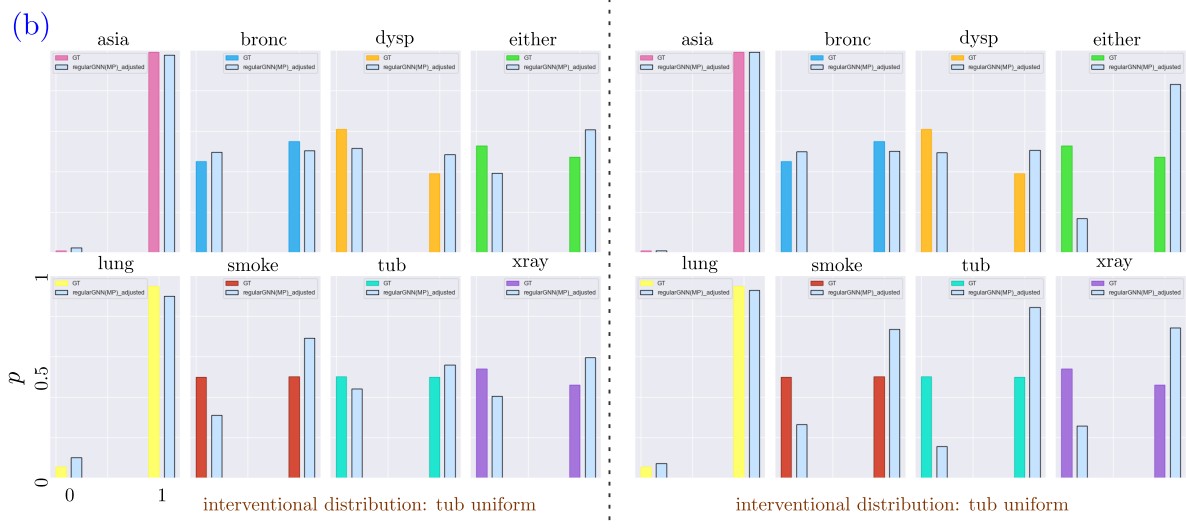

Figure 10: **Systematic Investigation: Question (b).** How does variance in ELBO during variational optimization affect the method? (Best viewed in color.)

**Q-(b) How does variance in ELBO during variational optimization affect the method?** Consider Fig.10 in the following. Two different random seeds (that is, different initializations and thus optimization

trajectories) for the same iVGAE under same settings (data, training time, etc.) are being shown. Clearly, the optimization for the seed illustrated on the left was successful in that the quantities of interest are being adequately estimated. However, the random seed shown on the right overestimates several variables (tub, either, xray) and simply does not fit as well. We argue that this is a general property of the variational method and ELBO (Eq.2) i.e., the optimization objective is non-convex and only a local optimum is guaranteed. Put differently, the variance in performance amongst random seeds (as measured by ELBO) is high.

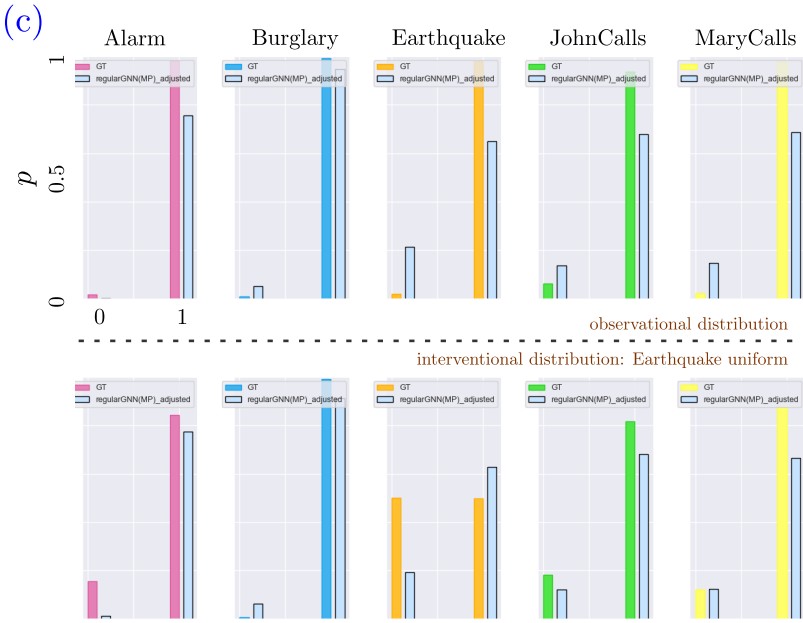

Figure 11: **Systematic Investigation: Question (c).** When and how does the method fail to capture interventional distributions? (Best viewed in color.)

**Q-(c) When and how does the method fail to capture interventional distributions?** Consider Fig.11 in the following. The predicted marginals of a single iVGAE model on the Earthquake dataset (Korb & Nicholson, 2010) are being presented for the observational density (top) and the interventional $do(\text{Earthquake} = \mathcal{B}(\frac{1}{2}))$ (bottom). The underlying graph in this real-world inspired data set is given by

$$G = (\mathbf{V}, \mathcal{E}) = (\{B, E, A, M, J\}, \{(B \to A), (E \to A), (A \to \{M, J\})\}), \tag{17}$$

where $B, E, A, M, J$ are 'Burglary', 'Earthquake', 'Alarm', 'MaryCalls' and 'JohnCalls' respectively. From $G$ we can deduce that the mutilated graph $G_I$ that is generated by the aforementioned Bernoulli-intervention $I := do(E = \mathcal{B}(\frac{1}{2}))$ will in fact be identical $G = G_I$. Put differently, conditioning and intervening are identical in this setting. The formulation for performing interventions in GNN (Def.1) only provides structural information i.e., information about the intervention location *but not about the content of the intervention.* While this generality is beneficial in terms of assumptions placed onto the model, it also restricts the model in this special case where associational and interventional distributions coincide. In a nutshell, computationally, the two posed queries $I_1 = I$ and $I_2 = do(\emptyset)$ are identical in this specific setting ($I_1 = I_2$) and this is also being confirmed by the empirical result in Fig.11 i.e., the predictions are the same across all settings as follows naturally from the formulation in Def.1 which in this case is a drawback. Generally, this insight needs to be considered a drawback of formulation Def.1 opposed to being a failure mode since the formulation indeed behaves as expected. In all our experiments, actual failure in capturing the densities seems to occur only in low model-capacity regimes, with early-stoppage or due to numerical instability.

**Q-(d) At what degree does the performance of the method vary for different training durations?** Consider Fig.12 in the following. It shows the same model being probed for its predictions of the observational

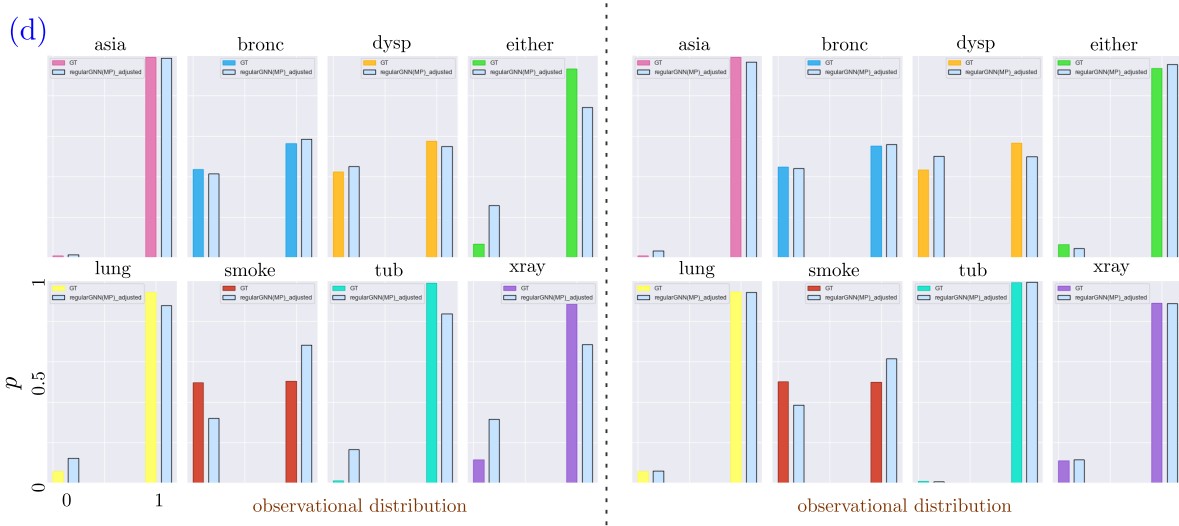

Figure 12: **Systematic Investigation: Question (d).** At what degree does the performance of the method vary for different training durations? (Best viewed in color.)

distributions at different time points, left is early and right is later (at convergence). Following intuition and expectation, training time does increase the performance of the model fit. Consider nodes 'tub' and 'lung' which were both underestimated in the earlier iterations while being perfectly fit later upon convergence.

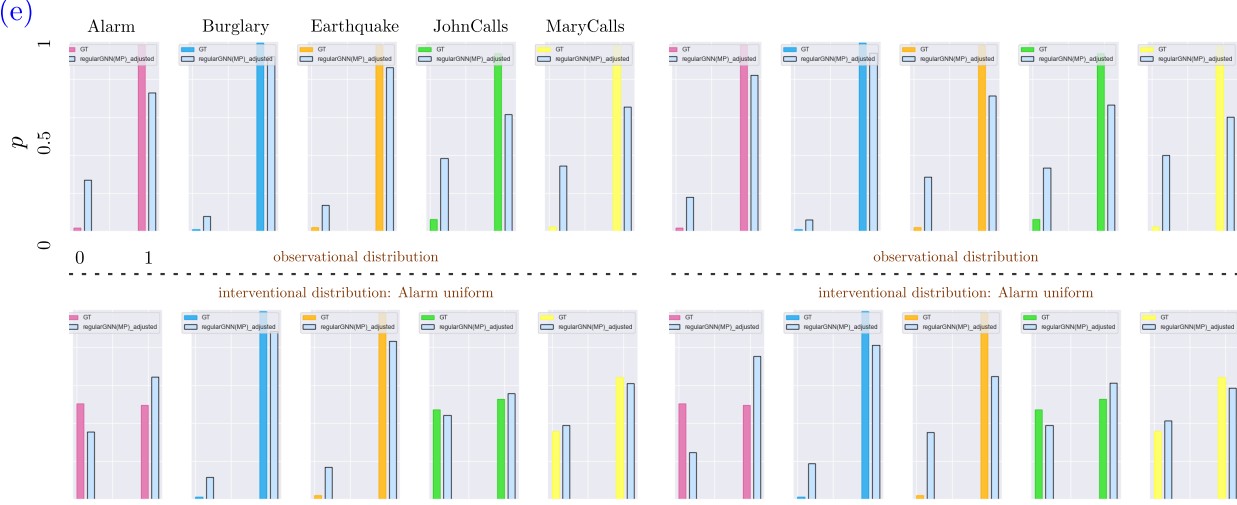

Figure 13: **Systematic Investigation: Question (e).** How does the method scale w.r.t. interventions when capacity is kept constant? (Best viewed in color.)

**Q-(e) How does the method scale w.r.t. interventions when capacity is kept constant?** Consider Fig.13 in the following. We show the same iVGAE model configurations being trained on either 2 interventional (left column) or 4 interventional distributions (right column) from the Earthquake dataset. I.e., we keep the model capacity and the experimental settings constant while increasing the difficulty of the learning/optimization problem by providing double the amount of distributions. As expected, we clearly see

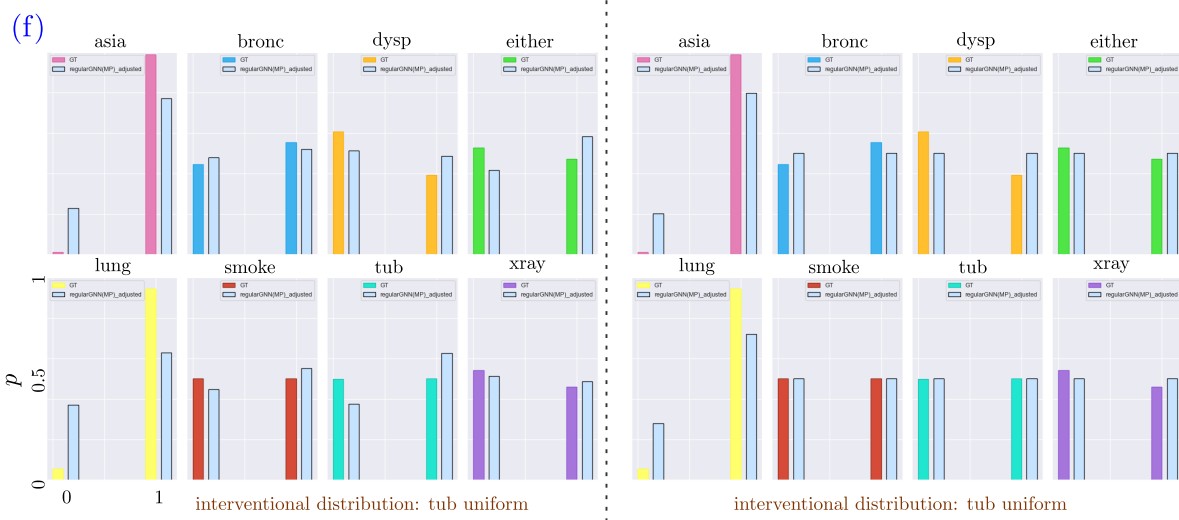

Figure 14: **Systematic Investigation: Question (f).** How important is parameter tuning? (Best viewed in color.)

a degeneration in the quality of density estimation. The iVGAE model trained on 2 distributions adequately estimates the distributions (lower left) while the model trained on more distributions fails (lower right).

**Q-(f) How important is parameter tuning?** Consider Fig.14 in the following. It shows an iVGAE model before and after parameter tuning (left and right respectively) on the Bernoulli-intervention $do(\text{tub} = \mathcal{B}(\frac{1}{2}))$ on the ASIA dataset where the tuned parameters involve aspects like pooling (sum, mean), layer numbers, learning rate, batch size etc. We clearly see an improvement towards a perfect fit for certain nodes ('smoke', 'tub', 'either'). As expected, parameter tuning, as for any other machine learning model, is essential for improving the predictive performance. Especially for universal density approximation this is crucial—since in principle any density can be approximated with sufficient capacity and thus is dependant on not only on the model itself but also on the optimization.

**Numerical Report.** In Tab.2 we show numerical statistics on the trained models applied to the different data sets for answering the investigated questions (a)-(f). For reproducibility and stability, we trained 10 random seeds per run. `NaN` values might occur for a single seed due to numerical instability in training, thus invalidating the whole run. We show the performance on different, increasing interventional data sets at various training iterations. We report mean, best and worst ELBO and log-likelihood performances (the higher the better). Question (b) regarding the variance of ELBO becomes evident when considering the best-worst gaps. As expected, ELBO lower bounds the marginal log-likelihood. Also, by providing more distributions to learn, thus increasing difficulty, the quality of the fits in terms of ELBO/likelihood degenerates which is inline with what we observed regarding question (e). Finally, we might also note that the validation performance, as desired, corresponds to the test performance.

## 5.2 Causal Effect Estimation

**TL;DR.** Fig.15 reveals that a PCM like iVGAE is capable of competing with reported results in the literature (e.g. NCM) when it comes to causal effect estimation (here ATE) *although not being an SCM*, as compared to the SCM the iVGAE is significantly compressed in terms of model complexity (only $\mathcal{O}(1)$ sub-models $n$ opposed to $\mathcal{O}(n)$ as for SCM/NCM).

Our experiments in the following investigate causal inference in the form of causal or treatment effect estimation. We are interested in the average treatment effect defined as $\text{ATE}(X, Y) := \mathbb{E}[Y \,|\, do(X{=}1)] -$

| Dataset | $|\mathcal{L}_2|$ | Steps | Mean Train ELBO | Mean Valid ELBO | Mean Test ELBO | Mean Valid $\log p(x)$ | Mean Test $\log p(x)$ | Best Test ELBO | Worst Test ELBO |
|---------|------|-------|-----------|-----------|----------|--------------|-------------|-----------|------------|
| ASIA | 2 | 16k | -3.43 | -6.02 | -4.60 | -4.15 | -4.11 | -4.10 | -5.37 |
| ASIA | 4 | 16k | NaN | NaN | -4.61 | NaN | -4.05 | -3.79 | -5.59 |
| Cancer | 2 | 12k | -2.26 | -4.76 | -3.17 | -3.66 | -2.76 | -2.35 | -4.49 |
| Cancer | 4 | 12k | NaN | NaN | -3.26 | NaN | -2.88 | -2.43 | -4.53 |
| Earthquake | 2 | 12k | -1.21 | -3.02 | -2.43 | -1.92 | -1.93 | -1.49 | -3.50 |
| Earthquake | 4 | 12k | -0.78 | -4.67 | -2.77 | -2.31 | -2.27 | -1.75 | -3.46 |

Table 2: **Causal Density Estimation: Key Statistics.** The aggregations cover 10 random seeds for each of the models respectively. Details in the main text.

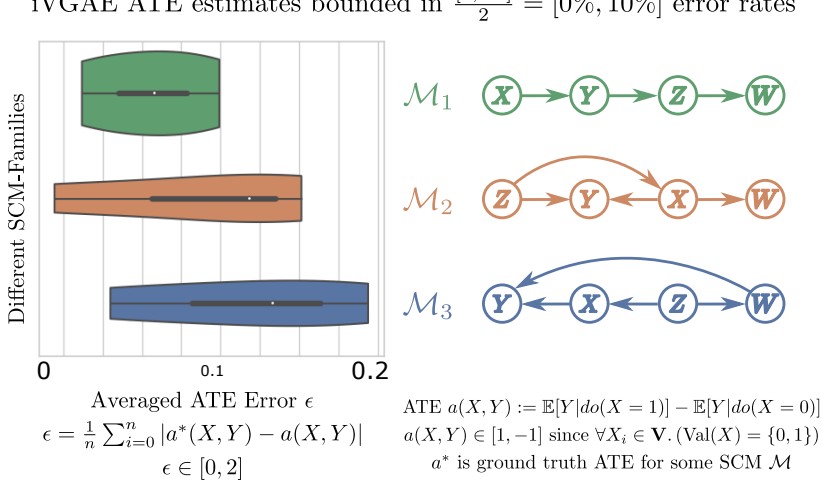

iVGAE ATE estimates bounded in $\frac{[0,0.2]}{2} = [0\%, 10\%]$ error rates

Different SCM-Families

Averaged ATE Error $\epsilon$
$\epsilon = \frac{1}{n}\sum_{i=0}^{n}|a^*(X,Y) - a(X,Y)|$
$\epsilon \in [0,2]$

ATE $a(X,Y) := \mathbb{E}[Y|do(X=1)] - \mathbb{E}[Y|do(X=0)]$
$a(X,Y) \in [1,-1]$ since $\forall X_i \in \mathbf{V}. (\text{Val}(X) = \{0,1\})$
$a^*$ is ground truth ATE for some SCM $\mathcal{M}$

Figure 15: **Causal Effect Estimation.** The iVGAE estimates the ATE on average with error rates up to maximally 10% on the three SCM families $\mathcal{M}_i$ under consideration. (Best viewed in color.)

$\mathbb{E}[Y \mid do(X{=}0)]$, where the binary variable $X$ is being referred to as treatment and $Y$ is simply the outcome variable or effect e.g. patient recovery. Note that we can extend the ATE to be categorical/continuous, however, we focus on binary structures in the following, thereby the aforementioned formula is sufficient. Also, note that ATE can also be viewed as a special case of density estimation in which the same intervention location $X$ is being queried for the different intervention parameterizations $do(X = x)$, for binary variables this amounts to $do(X = a), a \in \{0,1\}$. General properties of the density estimation for the iVGAE have been systematically investigated in the previous subsection (see main Fig.8 and Figs.9-14)). Effect estimation is an important sub-category of causal inference since usually one is interested in a *specific* causal relation rather than minuscule approximation of all possibly derivable probability densities within the causal realm. For instance, a medical doctor might only be interested in the consequences of administering the particular drugs available for patient's treatment. In the following we do not consider identification in the usual sense, since the iVGAE is a *data-dependent* model and not an SCM. An SCM (and by extension any NCM) is by construction capable of identification, for the iVGAE it is rather a special case of estimation again. In the following, Fig.15 will act as guiding reference to the subsequent analysis.

**Considered SCM Structures.** In the following we *always* consider the ATE of $X$ on $Y$, so the SCM structure are chosen relative to the pair $(X,Y)$. We consider 3 different SCM families (1: chain, 2: confounder, 3: backdoor) that are of significantly different nature in terms of information flow as dictated by the *d*-separation criterion (Koller & Friedman, 2009). Fig.15 portraits the implied graphs (note the re-ordering of the variables, the graphs are being drawn in a planar manner). In the following we provide the exact

| $\mathfrak{C}_i$ | $do(X = x)$ | Steps | Mean Train ELBO | Mean Valid ELBO | Mean Test ELBO | Mean Valid $\log p(x)$ | Mean Test $\log p(x)$ | Best Test ELBO | Worst Test ELBO |
|---|---|---|---|---|---|---|---|---|---|
| 1 | 1 | 2.5k | -1.48 | -1.60 | -1.59 | -1.43 | -1.43 | -1.50 | -1.86 |
| 1 | 0 | 2.5k | -1.86 | -2.12 | -2.07 | -1.85 | -1.84 | -1.70 | -2.47 |
| 2 | 1 | 2.5k | -1.48 | -1.60 | -1.59 | -1.43 | -1.43 | -1.37 | -1.86 |
| 2 | 0 | 2.5k | -1.86 | -2.12 | -2.07 | -1.85 | -1.84 | -1.91 | -2.47 |
| 3 | 1 | 2.5k | -1.48 | -1.60 | -1.59 | -1.43 | -1.43 | -1.37 | -1.86 |
| 3 | 0 | 2.5k | -1.86 | -2.12 | -2.07 | -1.85 | -1.84 | -1.91 | -2.47 |

Table 3: **Causal Effect Estimation: Key Statistics.** The aggregations cover three random seeds per model. Details in the main text.

parametric form for each of the SCMs $\mathcal{M}_i, i \in \{1, 2, 3\}$, first, the chain is given by

$$
\mathcal{M}_1 = \begin{cases}
X \leftarrow & f_X(U_X) = U_X \\
Y \leftarrow & f_Y(X, U_Y) = X \wedge U_Y \\
Z \leftarrow & f_Z(Y, U_Z) = Y \wedge U_Z \\
W \leftarrow & f_W(Z, U_W) = Z \wedge U_W,
\end{cases}
\tag{18}
$$

the confounded structure is given by

$$
\mathcal{M}_2 = \begin{cases}
X \leftarrow & f_X(Z, U_X) = Z \oplus U_X \\
Y \leftarrow & f_Y(X, Z, U_Y) = (X \wedge U_Y) \oplus (Z \wedge U_Y) \\
Z \leftarrow & f_Z(U_Z) = U_Z \\
W \leftarrow & f_W(X, U_W) = X \wedge U_W,
\end{cases}
\tag{19}
$$

and finally the backdoor structure is given by

$$
\mathcal{M}_3 = \begin{cases}
X \leftarrow & f_X(Z, U_X) = Z \oplus U_X \\
Y \leftarrow & f_Y(W, X, U_Y) = X \wedge (W \wedge U_Y) \\
Z \leftarrow & f_Z(U_Z) = U_Z \\
W \leftarrow & f_W(Z, U_W) = Z \wedge U_W,
\end{cases}
\tag{20}
$$

where $\oplus$ denotes the logical XOR operation. We use logical operations to assert that the variables remain within $\{0, 1\}$. Note that in this specific example we consider Markovian SCM, thus the exogenous variables are independent. We choose Bernoulli $\mathcal{B}(p), p \in [0, 1]$ distributions for parameterizing the exogenous variables. We choose the $p_i$ for each of the terms $U_i$ uniformly at random to generate 5 different parameterizations of the same structure. For each intervention we create a data set of size 10,000 and train a model consisting of two iVGAE modules. We consider 3 random seeds for each of the 3 parameterizations for each of the 3 structures, resulting in $3^3 = 27$ distinct optimizations. In the following we always consider the ATE of $X$ on $Y$, that is $Q := \text{ATE}(X, Y)$, which can be positive/negative with $Q \neq 0$ or neutral with $Q = 0$ if there is neither a direct nor indirect influence from $X$ to $Y$. All ATE estimates we observed were approaching zero. Specifically, the errors were bounded in $[0, 0.2]$ whereas the maximum possible error is $|\text{ATE}^* - \text{ATE}| = |1 - (-1)| = 2$ i.e., the worst-case single approximation was off by only 10%. Therefore we can argue the estimates to be competitive and thereby reasonable.

**Interpretation for ATE Estimation on the Chain SCM $\mathcal{M}_1$.** The causal effect of $X$ on $Y$ is both direct and unconfounded. It is arguably the easiest structure to optimize for and as expected the iVGAE performs adequately (see top/green row in Fig.15). The variance is further reduced in comparison to the other SCM families, arguably due to the relatively low variance in the ground truth ATEs (as $\mathcal{M}_1$ ATE are mostly positive).

**Interpretation for ATE Estimation on the Confounder SCM $\mathcal{M}_2$.** The causal effect of $X$ on $Y$ is direct, yet confounded via $Z$. The ATE can thus obtain positive, negative and also the zero value given a

particular parameterization. The iVGAE reacts accordingly and is able to adequately estimate the causal effect for any such parameterization. This is the key observation to assert valid causal inference since a *correlation-based model would fail* to return the correct answer i.e., it would simply return $p(Y|X)$ instead of $p(Y \mid do(X))$) which in this case would be incorrect since the causel effect is confounded.

**Interpretation for ATE Estimation on the Backdoor SCM $\mathcal{M}_3$.**   This family poses the conceptually most difficult case since the causal effect of $X$ on $Y$ is confounded through a backdoor path $X \leftarrow Z \cdots Y$. Nonetheless, the iVGAE prevails in adequately modelling the causal effect. The variance, like for $\mathcal{M}_2$, is increased which is arguably due to the spectrum of available ATE instantiations dependent on the concrete parameterization of the SCM.

**Numerical Report.**   In Tab.3 we show numerical statistics on the trained models applied to the different SCMs $\mathcal{M}_i$ for one of the 3 parameterizations averaged across 7 random seeds. We show the two interventions on $X$ for computing the ATE, and report mean, best and worst ELBO and log-likelihood performances (the higher the better). As expected, ELBO lower bounds the marginal log-likelihood and the validation performance, as desired, is in correspondence to the observed test performance.

## 6   Conclusions and Future Work

Our goal with this work was to get one step closer to an integration of causality with today's machine learning methodologies like neural networks. We wish to do so since both worlds offer amazing capabilities that seem crucial for any future agent capable of intelligent behavior. Ultimately, we have thus envisioned a fully-differentiable system that combines the benefits of both worlds. Thought provoking at first was the idea that graph neural networks and structural causal models might have more in common than previously thought. Following this gut feeling, we started our analysis with the common ground between GNN and SCM, namely (implied or assumed) *graphs.* By formalizing the key concept of *intervention* within GNN, we discovered three new model classes. First, the minimal NCM which hides away the structural equations of an SCM in its implicit message-passing that is in essence just a single GNN. Second, the maximal NCM which employs neural models for each causal relation pair separately. Finally, third, the first partially causal model based on GNN that we named iVGAE. To our surprise, as we did not anticipate this discovery until formalizing interventions in GNN and continuing from there, came the notion of *partial causal models* which seem to lie somewhere right in the middle of the spectrum of models spun by non-causal models and classical SCM at the extreme ends. These models are particularly interesting for ML since they allow to output correct, causal behavior where needed, while still being compact and easier to learn. To this end, the bulk of the discussion in this paper was spent in favor of iVGAE, the first GNN-based PCM. Our theoretical discussions proved conditions under which the uncovered model(s) are equivalent, feasible or expressive amongst others, while our empirical analysis corroborated for iVGAE specifically the established results underlining some technical details when actually learning such models in practice. Looping back to our original question at the beginning of the paper "If we take causality and somehow sensibly combine it with modern machine learning like deep learning, will it lead to (deep) *understanding*?", we believe that the evidence points to a 'Yes,' nonetheless, we are unable to give a definite answer and rather rephrase our conclusions to lie in the surprising discovery of different *shades* of NCM (i.e., there is no single, universal definition of NCM) and even a completely new *class* of causal models, namely PCM (i.e., causality in AI is not just a binary property of a model).

We hope that this present work will spark further research on the spectrum spun by non-causal and causal models, identifying practical causal inference for a tighter integration between AI and causality.

What we ourselves envision for future research is many-fold. We believe that an extension of the definition of an intervention on GNN might be valuable since the proposed notion is restricted to knowing only "that and where" an intervention occurred, which naturally restricts it from supporting the complete PCH. Ideally, we want to have fully-causal models that are not restricted in inference or learning like any form of parameterized SCM we know of, like any of the discussed NCM variants, are. We believe this to be sensible since it can be argued that in practice we are rarely interested in having *exactly* one particular SCM but rather we want to be correct in certain sub-components of the overarching system (similar to how causal effect estimation might only care about a quantity like the ATE and does not care about the actual structural equation). In a sense,

this is an argument of *abstraction* as causality itself can be seen as an abstraction from differential equations. On another note, concepts like *persistent message passing* (Strathmann et al., 2021) are interesting ideas from geometric deep learning that might allow for similar achievements along that direction by enabling an effective notion of counterfactuals within PCM. Naturally, testing on larger causal systems and adding conditions to iVGAE for counterfactuals would pose two pragmatic next steps which might generally, if successful, contribute to an increased interest in research around the integration of machine learning and causality. Most importantly though, we feel that the introduction of PCM is a foundational new idea that is worthwhile exploring as it trades off expressivity with complexity and it is not clear yet whether and how identification might occur in these models. As we have seen, iVGAE have identification and estimation essentially coincide, but that might not hold for general PCMs. Exploring what other PCMs exist and questioning whether maybe the definition of a PCM that we propose is still too restrictive to capture what truly makes a model *partially causal* are important directions of future research.

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

## A    Remaining Proofs

To improve readability of the main paper's content we provide the proofs following the one after Thm.1 in this appendix.

### A.1    Proof for Maximal NCM

We observe a special case of classical NCM with the following corollary.

**Corollary 1** (C2(ii), Maximal NCM)**.** *Assume the setting from Thm.1. Further, we allow for the violation of sharedness of $\psi$ by arbitrary parameterization with MLP. The resulting computation is an NCM special case with a computation per-edge.*

*Proof.* The proposed computation layer (Maximal NCM) is a special case of the classical NCM in the way that it represents the MLP as a linear combination of sub-MLPs (as in Thm.1 to the extent to which the original structural equation of the SCM is linearly decomposable). Thereby, we have that

$$\mathbf{h}_i = f_{ii}(U_i, \mathcal{A}_i) + \sum_j f_{ij}(V_j) = \sum_j \text{MLP}_j^{\boldsymbol{\theta}_j} = \text{MLP}_i^{\boldsymbol{\theta}} = f_i \tag{21}$$

where the first equality follows from Thm.1, the second equality from our assumption that $\psi$ can be replaced by arbitrary MLP, the third equality from $\text{MLP}_i^{\boldsymbol{\theta}} = \sum_k \text{MLP}_k^{\boldsymbol{\theta}_k}$ (Hornik et al., 1989) which gives the last equality being inline with the classical definition of NCM. □

### A.2    Proof for Interventional Equivalence

A well-defined intervention for GNN that behaves natural w.r.t. interventions in SCM. Hard interventions are considered where the parental dependencies are cut.

**Proposition 1.** *Let $\mathcal{M}$ be an SCM with graph $G$ and let $f$ be a GNN layer corresponding to $\mathcal{M}$. An intervention $do(\mathbf{X}), \mathbf{X} \subseteq V$, on both $\mathcal{M}$ and $f$ produces the same intervened graph.*

*Proof.* An interventional SCM $\mathcal{M}_{\mathbf{x}}$ is a submodel of the original SCM $\mathcal{M}$ where the structural equations for variables $\mathbf{X}$ are being replaced by the assignment $\mathbf{x}$. Through this operation, denoted by $do(\mathbf{X} = \mathbf{x})$, the dependency between the causal parents of any node $V_i \in \mathbf{X}$ is being lifted (as long as the assignment $\mathbf{x}$ is not dependent on the parents). Therefore the intervened graph is given by $G_{\mathbf{x}} = (\mathbf{V}, \mathcal{E} \setminus \{(j, i) \mid V_j \in \text{pa}_i, V_i \in \mathbf{X}\})$ with $G = (\mathbf{V}, \mathcal{E})$. Intervening on a GNN layer that uses $G$ implicitly considers a modified neighborhood $\mathcal{M}_i^G = \{j \mid j \in \mathcal{N}_i^G, j \notin \text{pa}_i \iff i \in \mathbf{X}\}$ which removes exactly the relations to the parents. Since the original graphs are the same, the intervened graphs must also be. □

### A.3    Proof for SCM Consistency with GNN-based PCM iVGAE

The proof is an analogue to Thm.1 in (Xia et al., 2021) but saves a lot in that a projection onto a notion of *canonical* SCM is not necessary because of the restriction on $\mathcal{L}_2$ as suggested by (Peters et al., 2017).

**Theorem 2.** *For any SCM $\mathcal{M}$ there exists an iVGAE $\mathcal{V}$ for which $\mathcal{V}$ is $\mathcal{L}_2$ consistent w.r.t. $\mathcal{M}$.*

*Proof.* Let $\mathcal{V}$ be an iVGAE and $\mathbf{D} = \bigcup_{i=1}^k \{\mathbf{D}_i\}$ a collection of data sets on the variables $\mathbf{V}$ of an arbitrary SCM $\mathcal{M}$ for multiple interventions $k \in \{2, \dots, n\}$. That is, for each data set it holds that $\mathbf{D}_i \sim p_i \in \mathcal{L}_j(\mathcal{M}), j \in \{1, 2\}$. Note that the observational case ($\mathcal{L}_1$) is considered to be an intervention on the empty set, $p(\mathbf{V} \mid do(\emptyset)) = p(\mathbf{V})$ therefore $\mathbf{D}$ contains at least the observational case and one intervention or two interventional cases. Since we know that there always exists a parameterization of the encoder-decoder pair of $\mathcal{V}$ such that any distribution $p$ can be modelled to an arbitrary precision[16], we have that $p^{\mathcal{V}} = p_i$. Since $k > 1$ we further have that $\mathcal{V}$ models the PCH up to level partially $\mathcal{L}_2(\mathcal{V})$. Finally, since the distributions are modelled relative to $\mathcal{M}$ (since $p_i \in \mathcal{L}_j(\mathcal{M})$), we have partial consistency $\mathcal{L}_2(\mathcal{V}) \subset \mathcal{L}_2(\mathcal{M})$. □

---

[16]This is a long standing result on universal approximators for densities (UDA; see Goodfellow et al. (2016); Plataniotis & Hatzinakos (2017)) which extends to i-/VGAE. Choosing a Gaussian Mixture as variational family is already sufficient for UDA.

### A.4 Proof for *Partial* Causal Hierarchy Theorem

Assuming the truth of the Causal Hierarchy Theorem (and it needs to be assumed since the proof of the original theorem was not made publically available by the authors, however, an alternate version of a CHT proof was recently compiled using topology (Ibeling & Icard, 2021), therefore, justifying this assumption), we can also provide a CHT result for the notion of partial consistency and in this case for our PCM iVGAE, both of which are introduced in the main paper.

**Corollary 2** (Partial-CHT for PCM). *Consider the sets of all SCM and PCM, $\Omega, \Upsilon$, respectively. If for all $\mathcal{V} \in \Upsilon$ it holds that $\exists q \subset \mathbb{N}. [\mathcal{L}_1(\mathcal{V}) = \mathcal{L}_1(\mathcal{M}) \implies \mathcal{L}_2^q(\mathcal{V}) = \mathcal{L}_2^q(\mathcal{M})]$ with $\mathcal{M} \in \Omega$ and $q$ choosing a subset of $\mathcal{L}_2$ distributions, then we say that layer 2 collapses relative to $\mathcal{M}$. By extension of the classical NCM result on the CHT, it is clear that on the Lebesgue measure over SCMs the subset in which layer 2 of iVGAE collapses to layer 1 has measure zero.*

*Proof.* The proof is unfortunately not an immediate consequence of prior results since PCM are *not* parameterized variants of SCM. However, we can make it work through the notion of *selectors* for our PCM-based distributions. And we can still re-use a prior result on classical NCM, while using the definition of SCM-collapse as given by (Bareinboim et al., 2020). To be specific, our $\mathcal{L}_i$ subset selection indexed by $q$ is always chosen such that $\mathcal{L}_2^q(\mathcal{M}) = \mathcal{L}_2^q(\mathcal{V})$ for SCM-PCM pair $(\mathcal{M}, \mathcal{V})$. If layer 2 collapses to layer 1 relative to $\mathcal{M}^*$ then any SCM $\mathcal{M}$ will have that $\mathcal{L}_1(\mathcal{M}) = \mathcal{L}_1(\mathcal{M}^*)$ and $\mathcal{L}_2^q(\mathcal{M}) = \mathcal{L}_2^q(\mathcal{M}^*)$. W.l.o.g., we will consider iVGAE as our PCM representative. By Thm.2 we know that there will always exist a corresponding iVGAE $\mathcal{V}$ that is $\mathcal{L}_2$-consistent with $\mathcal{M}$ but since it is consistent with $\mathcal{M}$ and not $\mathcal{M}^*$ it follows the same behavior that $\mathcal{L}_1(\mathcal{V}) = \mathcal{L}_1(\mathcal{M}^*)$ and $\mathcal{L}_2^q(\mathcal{V}) = \mathcal{L}_2^q(\mathcal{M}^*)$, which means that the layer 2 also collapses for the iVGAE model. The analogue argument holds in reverse when layer 2 does not SCM-collapse to layer 1 relative to $\mathcal{M}^*$. Since both directions together suggest an equivalence on the way collapse occurs for both SCM and iVGAE, we have the PCHT established. $\square$

