# OpenReview forum: "Interventions in Graph Neural Networks Lead to New Neural Causal Models"
_TMLR — Rejected by TMLR_

### Review · Reviewer_VTbv · 2023-05-06

**Summary Of Contributions:**

This paper explores an interesting topic related to neural causal models, i.e., to investigate whether one could introduce additional inductive bias into GNNs such that they can then be used for causal inference. The authors formulate a new GNN layer that take into account the causal graph in an SCM and shows (both theoretically and empirically) that such GNNs can behave reasonably when estimating conditional and interventional distributions as well as ATE. Overall, it is a good read and I feel that it would be above the bar for TMLR. I do have some comments and concerns, which are listed below.

**Audience:**

Yes

**Claims And Evidence:**

Yes

**Requested Changes:**

Better organization and formatting, especially in Section 3 (see details above).

Clarification on the overloading of variables such as $f_i$.

Def. 3 only guarantees that the model V can emit certain distributions. Does it also guarantee that the distributions emitted should be identical (in terms of parameters, e.g., mean and variance for Gaussians) to those emitted by M?

Discussion on the commonalities and differences between neural *causal* models (NCM), which is a tight integration of deep learning and structural causal models (SCMs), and *non-causal* Bayesian deep learning (BDL) (e.g., [1] or [2]), which is a tight integration of deep learning and Bayesian Networks.

Could you also provide more details on why the bounded conditions are needed in Def. 3?

Definition of $B(1/2)$ on Page 15 and 17.

More details on how ATE estimation is done in Section 5.2 and experiments for Q(a).

[1] A survey on Bayesian deep learning. ACM Computing Surveys 2020.

[2] Towards Bayesian deep learning: a framework and some existing methods. TKDE 2016.


**Strengths And Weaknesses:**

I appreciate the thorough notation descriptions and also like Fig. 5, which nicely summarizes the differences among SCM and different NCMs.

While the general idea of using VGAE to ‘simulate’ an SCM is not new (as it has been studied in the context of causal discovery), the relation between GNNs and SCMs is definitely an interesting topic, and this paper makes solid contributions in this direction.

In Eq (9), does it mean that even for an intervened variable $X$, the message passing will go from $X$’s children to $X$, since only parents $pa_i$ are excluded during message passing? Do you use a directed or undirected graph as $G$ in GNN? If former shouldn’t $N_i\setminus pa_i$ simply be an empty set?

In the statement of Theorem 1, it is a bit confusing that $h_i=f_i$, with $h_i$ as a “vector” and $f_i$ as a function. I assume the authors meant to overload the notation $f_i$ as a function and the result of a function?

The paper could have been better organized, especially within Section 3.

It might be helpful to discuss the commonalities and differences between neural *causal* models (NCM), which is a tight integration of deep learning and structural causal models (SCMs), and *non-causal* Bayesian deep learning (BDL) (e.g., [1] or [2]), which is a tight integration of deep learning and Bayesian Networks.

Def. 3 only guarantees that the model V can emit certain distributions. Does it also guarantee that the distributions emitted should be identical (in terms of parameters, e.g., mean and variance for Gaussians) to those emitted by M?

Could you also provide more details on why the bounded conditions are needed in Def. 3?

On Page 14, by batch size do you mean the number of variables or the ‘batch size’ as in neural networks’ SGM procedure.

Results on Q(a) look promising. For these experiments, what is the difference between their corresponding iVGAE models? Is it that the former is a simple VGAE while the latter is an iVGAE model constructed according to Eq (8) and (9)? Also, how are they trained? I assume the former is trained on observational data while the latter is trained on (synthetic) interventional data (using Bernoulli intervention)?

What is $B(1/2)$ on Page 15 and 17? It is used on Page 15 but not defined until on Page 17.

It is unclear how ATE estimation is done in Section 5.2. Do you train an iVGAE with the known SCMs using observational data, and then perform ATE estimation by randomly intervening X on the trained iVGAE and observe the effect on Y? Or is there a way to directly read out the ATE from a trained iVGAE?


Minor:

Page 6: Counterfactual/Interventional variables like $C_b$ in “the former $p(A_b, …, C_b)$” needs to be introduced before this sentence. Also $C_b$ seems to come out of nowhere.

The formatting is a bit strange, with very narrow white space between paragraphs. See Page 8’s first 2 paragraphs as an example.

Some text in Figure 7 is too small and not readable.

Typo on Page 11: causal cause -> causal because

Typo on Pag 12: before we now define…

Typo on Page 13: we there will always be -> there will always be

[1] A survey on Bayesian deep learning. ACM Computing Surveys 2020.

[2] Towards Bayesian deep learning: a framework and some existing methods. TKDE 2016.

---

> ### Author Response · Authors · 2023-05-08
>
> We would like to thank you, Reviewer VTbv, for the positive recommendation and all the insightful and detailed comments!
>
> Question regarding Eq.9: We opted for seeing the deletion of the parental relation as the defining property of a hard intervention. Therefore, the definition the way it is, which indeed in a undirected graph would allow for messages from the children of the intervened node. However, that would not be a causal graph (since causal models need directionality, even for cycles that holds true, we'd just have both directions simultaneously).
>
> Question regarding notation in Thm.: you are correct, we just overloaded that notation (took it as implict), we can add a supporting comment.
>
> The suggestion with BDL seems reasonable since BNs are historically speaking the predecessor to SCMs, although the discussion will just be a corollary essentially to the difference between (Causal) BNs and SCMs. A paragraph on this might be both an easy and efficient add-on.
>
> Question regarding Def.3: A consistency is actually enforced through the operator $\mathcal{L}_i(\mathcal{M})$ which gives the set of all distributions and the PCM need only a subset of that specific set. This should make it clear, but the consistency is a crucial point since otherwise, we'd have only a PCM and not a SCM-corresponding PCM, which is the defining property we (seek to) formalize. Also, for the bounds in the conditions, they make sure that our PCM does not turn out to be that very SCM i.e., it should indeed be a partially and not fully causal model.
>
> Question regarding batch size: correct, in the usual neural networks sense.
>
> Question regarding Q(a): indeed, this is just an iVGAE model and it validates that it correctly learned to predict the interventional distribution.
>
> Question regarding $B(\frac{1}{2})$ in the figure: this is just a fair coin flip, B should stand for Bernoulli here, like defined on p.17. We should put in the figure immediately as well.
>
> Question regarding ATE: we consider binary variables and therefore consider the full set of interventions (yes/no) and this also simplifies the ATE which end up being one probability term.
>
> Regarding $C_b$: it should be clear from context and then in the subsequent sentences it is being explained in more detail, was minor anyhow. One could even consider removing it, since counterfactuals are not as important for this work (at least the formalism).
>
> Thanks for recognizing the typos, we fix them immediately!
>
> Thanks again for the great feedback, we will make the changes right away!
>
> As soon as we've completed the requested changes, we will post the status update on here via another message.
>
> Your authors

---

> > ### Author Response · Authors · 2023-06-14
> > **Thank you for the comments**
> >
> > Dear Reviewer,
> >
> > We have now incorporated all the changes from your end in the new version of the paper. You can find the changes in **orange** in the newly uploaded pdf. We would like to thank you again as your comments helped in making the paper better.

---

### Review · Reviewer_YJuk · 2023-05-22

**Summary Of Contributions:**

The authors highlight connections between structural causal models and graph neural networks. The authors propose an equivalence between set of all GNN models and SCM's. Based on this, the authors propose two theoretical classes of GNN's, and one latent variable model that aims to learn the interventional distributions on a subset of variables in a dataset. The authors test their method on a number of datasets.

**Audience:**

Yes

**Broader Impact Concerns:**

To the best of my estimation this paper evokes no such concerns.

**Claims And Evidence:**

No

**Requested Changes:**

# Comments/questions/observations
- Pg. 1: I find the paper starting with a discussion regarding "deep understanding" unsupported and confusing. The authors might be served better by a more concrete framing of their work, even in the introduction.
- Pg. 1: "Thereby, combining causality with deep learning is of critical importance for research on the verge to a human-level intelligence." I do not think "combining causality with deep learning" is in and of itself specific enough to be desirable. The authors might want to be more specific about the kind of combination they are targeting.
- Pg. 1: I do not understand why the authors feel the need to define "neural causal model"s. That a neural network is used for any kind of learning task is not of interest in and of itself. Why not directly focus on the connection the authors are proposing? Also, although the authors explicitly mention that they do not define this rigorously, in future pages it is compared to SCM's and GNN's as a modeling framework.
- Pg. 2: "under which conditions machine learning is capable of causal reasoning ..." I do not exactly understand what this sentence implies.
- Pg. 2: I would urge the authors to be clearer about what PCM is from the get-go. The discussion and the authors exposition gets very confusing very quickly from the start of the paper.
- Pg. 2: "we show how GNN can be used to perform causal computations" - to the best of my knowledge causal computation is not a well defined term in terms of causality research. It would be more helpful to the reader to see exactly what kind of causality-related tasks GNN's are led to accomplish.
- Pg. 3: It is unclear how the latent variables in probabilistic models are "similar to disentanglement or causality". Please specify or omit.
- Pg. 3: How is VI overcoming curse of dimensionality?
- Pg. 4: The authors state that importance sampling based marginal likelihood estimation presents a "practical method for marginal inference" while citing Kingma and Welling 2019, yet the original paper states that "The downside of these approaches for optimizing a tighter bound, is that importance weighted estimates have notoriously bad scaling properties to high-dimensional latent spaces."
- Pg. 4: Please introduce the concept of endogenous and exogenous variables.
- Pg. 5:  On (6), is $p^{\mathcal{M}}(\mathbf{v}|do(\mathbf{W}))$ a typo?
- Pg. 5: "...distribution denoted $p^{\mathcal{M}}$": $p^{\mathcal{M}}$ has been used before, it might make sense to introduce it as the observational distribution first, explain do modifications later.
- Pg. 6: Please explain "Markovian SCM" explicity, even if briefly.
- Pg. 6: Please introduce the notation $p(\mathbf{A}_\mathbf{b})$ and $p(\mathbf{A}_\mathbf{b}, \dots, \mathbf{C}_\mathbf{b})$. Without doing so, (7) is indecipherable without explicit prior knowledge. If you assume this to be known do not introduce and direct the reader to appendix or an outside source.
- Pg. 6: What is "hard intervention"? Is it atomic intervention?
- Pg. 7: The paper seems to suffer from repeating introductory / justification sentences almost every few paragraphs. Still having not introduced their concepts, problem context, or results clearly, an extended discussion about whether interventions are important for their results is very hard to interpret, as it is still not clear what intervention exactly means in their translation of this concept to GNN's.
- Pg. 7: The authors reintroduce/relate SCM's, CBN's, FBN's, which they already discussed. Please consolidate these separate discussions.
- Pg. 7: In (9), does $\mathcal{N}^G_i$ the children of vertex $i$? I am assuming then even though the edges are directed, edge direction does not matter for being included in the neighborhood?
- Pg. 8: Again, the authors reiterate that intervention belongs on ladder 2 of PCH (already extensively covered).
- Pg. 8: The authors express their motivation again between Definition 1 and Theorem 1. Please consolidate discussions about the motivation of the work at the beginning of the paper.
- Pg. 8:  The preview before Theorem 1 does little to help the reader as it is still not clear what partial causality is or what a maximally parametrized NCM corresponds to, especially authors having expressed that NCM is not a strictly defined concept. I think it would be desirable if the authors simply proceeded to present their results.
- Pg. 8: In Theoremm 1, what does "similarity of SCM and GNN" refer to? I wanted to make sure as the theorem aims to show equivalence between SCM's and GNN's.
- Pg. 8:  How does the size of $\mathcal{A}_i$ affect the "similarity" of SCM and GNN? Is $\mathcal{A}_i$ not about only the non linearly decomposable effects of $pa_i$ on $i$? So regardless of the size of $\mathcal{A}_i$, the equivalent of message passing might or might not take place based on the linear partial effects?
- Pg. 9:  In (16), would it be correct to say $f_{XX}(U_X, \{Z\}):=f_{U_X}(U_X, Z) = \dots$? If so, it might be a good idea to make this explicit as this example is almost a "tutorial" to the conversion you introduced in Thm 1.
- Pg. 9:  What does "actual decomposition" refer to in Example 1?
- Pg. 10:  "A more radical view..." Are you positing an isomorphism between the sets of all SCM's and GNN's? If not, please clarify. If so, instead of a view this is a mathematical claim and needs to be explicitly stated in the theorem.
- Pg. 10: When you use NCM alone do you mean minimal NCM? Please clarify, and make sure to explicitly refer as it is quite hard to follow. I got this idea after reading post-Corollary 1, but Figure 5 suggests otherwise.
- Pg. 10: I fail to understand the distinction between minimal NCM's and maximal NCM's in any practical sense. Minimal NCM is learning separate parameters for each variable pairs as well, so in what particular way they are legimitately different than maximal NCM's? Given the confusion about these concepts, that the authors leave this discussion without any detailed discussion about the implications of such model classes to move on to a specific latent variable model leads the reader to fail to understand the significance of the authors' proposed model classes here.
- Pg. 11: This is an important example of the conceptual confusion that the paper invites at times.  "spectrum from noncausal models such as linear regression etc." I fail to understand the characterization of "(non)causal" model here. I would appreciate being a bit more specificity. As far as I can tell there's nothing inherent in a linear regression nor a CNN that makes them "causal" or "noncausal". Given there's no latent confounding or any other model mismatch, a linear regression can well be used to extract causal information. I do not understand what the scale of "causalness" of a model class should refer to, and I think the language needs to be much more exact in a paper that is explicitly about classes of causal models.





# Minor points, typos
- Pg. 1: What does "leveraging the exposed" mean?
- Pg. 1: "Any of the previously..." Typo?
- Pg. 2: Inductive bias is an existing term
- Pg. 2: "To link back to causality... " This sentence sounds like it proposes SCM is at the core of GNN's, not causality.
- Pg. 2: "To be more precise on the term causal inference" The previous sentences did not include the term "causal inference". It included related terms but causal inference has a specific meaning and should not be used arbitrarily. Moreover the following sentences do not really clearly explain the connection.
- Pg. 4: "sampling ... brings about sampling"
- Pg. 6: "common confounder" sounds confusing, "observed confounder" perhaps?

**Strengths And Weaknesses:**

Strengths: I believe that the potential connections between SCM's and GNN's the authors point out are interesting, and are worthy of further investigation. Indeed, such work has the potential of both further utilization of deep learning methods in causal analysis, as well as a more nuanced understanding of how such methods perform as they do, by analyzing their implicit inductive biases, some of which might have causal implications.

Weaknesses: I find that there are three large weaknesses of the paper that leads me to believe it is not ready to be published in its current state:

1- Fundamental conceptual issues with respect to the problem formulation and solutions offered,
2- The paper being unclear and/or inconsistent about what it promises,
3- The writing being disorganized and argumentation unclear.

The last point might have exacerbated the first two, since it was a very difficult read throughout, due to conceptual and organizational issues. I do like the authors' original idea of exploiting potential connections between GNN's and SCM's, and therefore provide much more detailed feedback below, in the hopes of understanding the authors' contributions better after their feedback.

---

> ### Author Response · Authors · 2023-06-14
> **Official Comment by Authors (1/2)**
>
> We thank the reviewer YJuk for the detailed comments/questions/observations in the "requested changes" section which helps us in  improving our paper. We now answer your comments here and all the required changes can be found in the newly uploaded pdf in **purple**.
>
> Regarding "deep understanding": clearly, this should be understood as a motivating vision (in a sense, the question whether causality is indeed what is needed for the next generation of learners) and is also supported by visions such as the ones by Pearl who uses the exact same wording "deep understanding." Can you re-specify more concretely what you feel is off, or does the now provided context lift the previous confusion? We've added a foot note on this in the beginning.
>
> Regarding NCM definition: since prior work to ours have proposed such models, however, have based their definition on particular design choices (in no way general), we do think it is important to define this. Would you agree?
>
> Regarding the PCM confusion: could you please be more specific, what exactly confused you about the prior discussion before we introduced our definition? To maybe give a short recap: PCM are a new class of models that we identify that sit between non-causal models and SCM.
>
> Regarding "causal computation" with GNN: it should simply understood as "causal model" that does "computation" equals "causal computation." We've added this to the paper.
>
> Regarding similarity to disentanglement and causality: the answer lies in the second half of the sentence, namely that the meaning of latent variables in latent variable models lies in them being some sort of "underlying, generative factors" for our data. This is clearly the interpretation of SCMs from causality and in disentanglement we usually try the same, find factors that are separate and explanatory of our data. We've added this to the paper.
>
> Regarding VI and curse of dimensionality: we refer for a proper account to the papers mentioned (Jordan et al., 1999; Blei et al., 2017) but on an intuitive level we make intractable queries tractable (or at least approximative) by using optimization. In reality, we really just re-formulate the problem to something that is just more practical. We've added this to the paper.
>
> Regarding importance sampling for marginal inference: we don't run into the issue of scaling in our experiments and the approach is a straightforward way to do marginal inference as suggested. We should add a footnote suggesting that this is not recommended though for scaling a system, for those interested in said direction. We've added this to the paper.
>
> Regarding exogenous/endogenous variables: these are standard terms in causality. Also it just means "inside" and "outside" system variables. Furthermore, we do have a footnote on the latter since they have a different interpretation to "regular" variables in the ML sense for instance. We've added this to the paper.
>
> Regarding (6): yes, this is a typo. Thanks, fixed.
>
> Regarding $p^{\mathcal{M}}$: indeed, in an introductory text-book it would make sense to introduce the observational before the interventional, but then again since this is the base level of understanding for causality and many other authors in the literature (e.g. Bareinboim or Peters et al.) use this notation, we had this as our base assumption for reading the paper. We've added it now.
>
> * Since some of these base ideas, which seemingly need explanation, is a re-occuring theme. The following list covers them. We've added all of these to the paper as requested:
>       1. Non-Markovian SCM simply refers to an SCM where endogenous variables can share exogenous variables. I.e., we have hidden confounders. Markovian SCM then means no hidden confounders.
>       2. For $p(\mathbf{A}_{\mathbf{b}})$ style notation for counterfactual "worlds" we refer to Bareinboim et al. "On Pearl's Causal Hierarchy" in addition to what we have written in the paper.
>       3. Hard intervention = atomic intervention = putting the value of an endogenous variable to a scalar, denoted $do(X=a), a\in \mathbb{R}$.
>       4. $\mathcal{N}$ means neighborhood and then sub-super-script denote for which node in which graph.
>
> Regarding re-iteration of interventions being on PCH: it seems that some basic concepts (like interventions) are not covered detailled enough but then the same concept, when re-iterated upon briefly in a later section, is considered as a sort of nuisance and should be cut. This seems contradictory. Thereby, we beg to differ on this point.

---

> > ### Author Response · Authors · 2023-06-14
> > **Official Comment by Authors (2/2)**
> >
> > Regarding "similarity of SCM to GNN" in Thm.1: this is being explained in that very paragraph in that $\mathcal{A}$ tells you something about how the computations in the GNN can be mapped to the causal effects in the SCM.
> >
> > Regarding your proposition of $f_{XX}(U_X,Z):=\dots$: no, since it is $U_X$ and not $X$.
> >
> > Regarding "actual decomposition": this simply refers to the "actual" $\mathcal{A}$ that we will encounter when looking at the conversion as in Thm.1, that is, for your specific SCM-GNN pair under consideration. We've added this to the paper.
> >
> > Regarding "a more radical view": we wrote this part because indeed our Thm.1 shows a correspondence between the sets of SCM and the sets of GNN, however, there is a gap between just having a computational model and the "right" model. SCMs are usually confused to be the "underlying reality," although they are simply a formalism for talking about causality. Any given SCM could potentially be the "underlying reality" for some phenomenon but it is not a necessity. We've added this to the paper.
> >
> > Regarding when writing NCM alone: we just mean the general case then, that is, either minimal or maximal since both of them compromise the set of NCMs. We specify minimal/maximal only when needed. We've added this.
> >
> > Regarding distinction of minimal/maximal NCM: the distinction should be simple, the former has minimal parameters (i.e., uses less neural networks) and the latter has more parameters (i.e., uses more neural networks). None of those were previously defined/discovered and thus pose original contribution. To make more clear how this fits into existing literature, we've add a separate paragraph in the paper.
> >
> > Regarding the confusion on "spectrum of (...)": of course models/techniques such as linear regression are used for causal inference, however, these models/techniques are themselves non-causal in the sense that we need assumptions outside the data and model to infer the causation. However, a SCM is clearly causal. It is defined as being the model that models causal relations. We've added a short paragraph on this.
> >
> > We look forward to discussing with you, thanks

---

### Review · Reviewer_dA5Y · 2023-06-01

**Summary Of Contributions:**

Authors propose using GNNs in neural causal models. They provide a set of definitions to better connect SCMs with GNNs both in observational and interventional layers of the hierarchy.

**Audience:**

No

**Broader Impact Concerns:**

Acceptance of this paper in its current form will be detrimental to the field of causality since it conveys a very confusing message about the role of GNNs in causality as it contains many unjustified claims to make GNNs relevant. The better way to do this in my opinion would be to find a very suitable dataset and solve a causal inference task using GNNs that cannot be solved with other deep learning architectures.

**Claims And Evidence:**

No

**Requested Changes:**

I am not sure I understand the goal of this paper. The theoretical results seem trivial. Interventions on an SCM - however it is parameterized via GNN or any neural net - clearly do create new SCMs. I am not sure why we need a new paper to say this.

Several vague claims beyond scientific are made in the paper without any justification. Some examples are:
"GNN-based causal inference that is necessary and sufficient for causal effect identification"

"and shows to be promising towards the dream of a system that performs causal inferences at the same scale of effectiveness as modern-day neural modules in their most impressive applications."

The contribution of the paper starts on page 7, so the first 7 pages can be significantly shortened.

Definition 1 simply removes parents of the intervened variables consistent with the definition of do interventions.

To be able to make sense of the asymmetry in causality, they also implicitly modify the GNNs to be unidirection, which I believe is not that common. This important point is only mentioned in passing in the last paragraph of page 7.

I do not understand the point of Theorem 1. Any GNN can be viewed as some SCM? Why is this statement interesting/useful?

Something called causal sub-effects is mentioned and used in the proof of Theorem 1 without a definition.

The authors "decompose" f but this is basically adding and subtracting the linear terms f_ij(V_j) where V_j belong to the parent set of node i. In general, these linear terms won't exist since the function is not decomposable. I think it would be better if the authors explicitly made the linear decomposability assumption rather than trying to operate in the most general setting at the cost of being implicit about the model/assumptions.

How does f_{ij} corresponds to a message computation if A=pa_i? If A=pa_i, then the function is not decomposable and f_{i,j} should be zero. I am not sure if I understand the reasoning here.

Are d_i, d_j used interchangeably with Vi, Vj in (10)-(12)? This is not clear.

In (12), the function \psi simply ignores one of the inputs. I am not sure why this is a good way to establish a correspondance between GNNs and SCMs. This feels forced to fit into the narrative of the authors.

Regarding the discussion after theorem 1: How does theorem 1 establish the connection between SCMs and GNNs if \psi is identical for all nodes but f is different for every node in the SCM, as discussed? I am not able to follow how these are not contradictory statements.

I also do not see how focusing on GNNs gives us any insight about the connection between SCMs and neural SCMs. One can trivially mimic the structure of the causal graph using neural networks as is done several times in the literature. Why does focusing on the subclass of GNNs give us any more insights?

"They are also the maximal NCM model class"
Why is this true? I believe this maximality claim needs a proof.

" since now we can use local messages parameterized by MLP that allow for computation per-variable"
I don't think this is established, is it? I missed the discussion where reusing was possible. In arbitrary SCMs, it is very unlikely that there will be room to reuse any computation. What are some practical settings where such reuse of computation could be relevant?

"In the special (or extreme) case of the original SCM being linear, the maximal NCM portrayed in Cor.1 is then maximally more fine-grained than the the definition of classical NCM since the former will model each of the causal sub-effects fij with separate MLPs, whereas the original NCM will choose only the structural equations to be modelled by MLPs i.e., the NCM is an aggregated or consolidated view of the maximal NCM."
It is very hard to parse such long sentences. What does it mean to be "maximally more fine-grained"?

I think the notions of minimal and maximal need to be defined. I am not sure what they mean in this context. Most discussions are very hand-wavy and lack any scientific rigor.

In section 4: the notion of partial causality does not make sense to me. What does that mean? Linear regression is called "non-causal". Why? Clearly, there are SCMs that are equivalent to linear regression. Not all of them are. Just like not all of them can be represented by GNNs. So I don't see the point of this distinction.

"they induce some causal inference capabilities"
What does it mean to induce causal inference capabilities?

"we observe that they are parameterized variants of an SCM and thus fully causal
models. "
I don't see this. The function re-using structure, and the symmetric structure of GNNs mean they are only a special case of SCMs. How are they "fully causal models" - and what does it mean to be  fully causal?

What is G in Definition 2? How is it tied to the GNN?

Proposition 1 is obvious because an intervention on a GNN is defined to mimic the way it acts on an SCM. In light of this, I don't find it insightful or useful in any context.

the word "dual" should not be used so hand-wavily since it means something specific in optimization.

Definition 3 is not explained fully. Why is the size of the interventional/counterfactual queries relevant in this conversation? What if the observed variables are discrete or continuous? The number of possible interventional queries even will be infinite (actually uncountable) then.

"These conditions are necessary conditions for these models to be considered partial as otherwise it would be either an standard SCM"
I am not sure if I understand why these conditions are necessary. What does "necessary" mean in this context?

"causal inference therefore still “makes sense” in that the different queries we can ask across the different Li are indeed qualitatively different from each other"
This is another vague and unscientific argument. What does it mean to "make sense" here?

The discussion between the model "complexities" of SCM vs iVGAE is very vague and does not lead to a nontrivial discussion at the end. What can one do with these observations?

It is not clear why we need to invoke GNNs or even deep learning in these small and low-dimensional datasets. It would be interesting if the authors demonstrated that the inductive bias of GNNs allows one to do better causal inferences in graph-structured datasets, like features defined on molecules.

It is not surprising to me that GNN layers can be used to model the SCMs, both theoretically and experimentally. The paper unfortunately fails to demonstrate interesting theoretical or experimental results.

Some Typos:
"parital"
Typical notation for post-interventional graph is G_{\bar{x}} not G_x.
"This theorem suggests that we there will always "
arguemnts->arguments in page 9

**Strengths And Weaknesses:**

Please see below for my detailed comments and concerns. In short, the paper contains a number of vague and unscientific claims. The contribution is not clear. None of the theorems contain any nontrivial results among them. Experiments are very limited to show the power of GNNs in this context.

---

> ### Author Response · Authors · 2023-06-14
> **Official Comment by Authors (1/2)**
>
> We thank reviewer dA5Y for the detailed comments, we wish to push back on many of these comments in the following argumentation, in hope of converging to a better paper together. The proposed changes are in **blue**  in the newly uploaded pdf.
>
> Interventions on SCMs create only modified SCMs that are up to the intervention the same as the base SCM. While this is not even the topic of our work, a discussion of parameterization, architecture etc. is highly non-trivial. With the same argument you could discredit all of Deep Learning. Therefore, we beg to push back on this statement.
>
> The statement "GNN-based causal inference that is necessary and sufficient for causal effect identification" has clear meaning in that we intend on using GNN models for performing causal inference, and we try to identify/characterize the criteria upon which we can indeed claim to have performed a valid causal inference. While not focusing much on effect identification, we focus on the models level of abstraction. We've added an improved comment to the paper.
>
> We disagree with the shortening of first 7 pages aspect since (a) page limits are not an issue (an informed reader can skip the said pages) and (b) that would be a general critique not of our paper but paper writing as a whole where "related work", "background" are standard and (c) a paper should be **self-contained** and since we use many different techniques/ideas (VI, Causality, Geometric DL), a discussion is necessary. It is a little surprising to get such a comment in a journal.
>
> Thm.1 is especially useful because it shows you, under consideration of $\mathcal{A}$, how we can map between computations in a GNN and causal effects in an SCM. This $\mathcal{A}$ will be very different for different structural equations of (different) SCMs. Furthermore, it raises the point of differentiation between "purely computational" and "causal" (or "true") model. We've added another comment on this to make this even clearer.
>
> Causal sub-effects are defined, both in said paragraph and the accompanying footnote. Since the term should be clear from that, a "big" definition (with a LaTeX definition environment) should not be necessary. We beg to differ here.
>
> The $\mathcal{A}$ term is a construction for handling exactly the non-linear cases. Therefore, the overall argument still holds. Of course, the linear SCM case is a nice special case of that. We've added more on this in the paper.
>
> Regarding $\mathbf{d}_i, V_i$: the former is GNN notation to denote feature vector of node $i$ while the latter is simply node $i$.
>
> Regarding (12): this is simply an artefact of the notations used in GNN and SCM literature. The causal sub-effect (or coefficient in a linear model) is a function only of that parent cause, whereas $\psi$ is a general computation rule and therefore also taking the actual variable under consideration as an argument. We've added this to the paper.
>
> Regarding GNNs and NCM variants: the distinction between minimal and maximal NCM is new and not part of prior literature. This came about while researching the relation between GNNs and SCMs, and in particular because of Thm.1, where for the proof to work we needed to find the correspondence of computations in the GNN to something in the SCM. That revealed more fine-grained useful parameterizations of NCM like for the maximal NCM. Further points on the minimal/maximal NCM related things:
>
> * Regarding proof for maximal NCM: we can add it as an explicit proof, but in fact this is what we considered "trivial" since there is no further decomposition of an SCM (beyond the SCM's causal sub-effects) to consider for the maximal NCM.
>
> * Minimal and Maximal NCMs are defined in Thm.1 and Cor.1 respectively, to repeat in simple terms: the minimal is what you get when you take the "classical" SCM and use even less neural modules, instead of using one for each structural equations you now use one for *all* structural equations, whereas for the maximal it is the opposite, you now use even one module for each of the parents within a structural equation.
>
> There are more details on this now in the paper.

---

> > ### Author Response · Authors · 2023-06-14
> > **Official Comment by Authors (2/2)**
> >
> > Regarding partial causality and Def.3: since this is a new notion that we introduce, we compiled a figure for better understanding (Fig 6 in the new pdf). Sure, you are right that some SCMs will represent linear equations, but they have clearly causal semantics (the causal relations are simply linear then), whereas just a linear model does not have that. I.e., only an SCM can talk about the ladder of causation not a linear model. However, this is nothing new yet. What is new is that our perspective on GNNs has allowed recognizing that there are models between SCM and non-causal models, that model only parts of the hierarchy. This is also what you see schematically depicted in the Figure we've shared here. We've added all of this to the paper.
> >
> > Regarding the wording "fully causal" that just means that we fully capture the hierarchy i.e., you have an SCM essentially. That is, "fully causal" = capable of generating all 3 types of distributions.
> >
> > Regarding $G$ in Def.2: that is simply the graph that defines the GNN computation, whereas $G_{\mathcal{M}}$ is the graph implied by the SCM $\mathcal{M}$. We've made this more clear in the paper.
> >
> > Regarding Prop.1 being "obvious": sure, one can make the argument but then again in the same way one could've made the argument above for minimal/maximal NCM. It is a short result that does not waste much space and is justified in our view in that it clearly demonstrates that interventions happen like in the usual, which confirms that we have a sensible definition for them. Therefore, we beg to differ on this point.
> >
> > Regarding the usage of the word "dual": (a) we use it only once in the whole paper in the context of Prop.1 so it is definitely not hand-wavy and (b) terms are being overloaded in all kinds of literature, therefore, this shouldn't be an argument. Furthermore, the dual here should simply refer to the two perspectives of GNN and SCM interventions which seems like a very fair use of the word on our side. Still, as it can only improve the paper, we've added this to the main paper.
> >
> > Regarding the phrase 'causal inference still "makes sense"': actually, this is a phrase encountered in discussions around the Causal Hierarchy Theorem (consider literature by Bareinboim et al. or talks by Judea Pearl on the topic) and furthermore we put it into quotations "makes sense" to highlight that. Really, what it tells you on a technical level is that inter-level inferences are impossible without assumptions. We've added this to the paper.
> >
> > Regarding the complexities: we should summarize this in a table to represent it better, thanks. But really what it tells you is that SCM can model L1, L2, L3, whereas iVGAE might only model L1, L2, and the former scales with the number of variables whereas the latter is actually constant! So it is fair to say it is a short discussion, but far from trivial. We've added this in the form of a new table to the paper.
> >
> > Regarding the empirical part: of course, we agree with you that deep learning does not show its impressiveness in small data sets, however, remember how this is not the point of Sec.5. Quite to the contrary, here we are concerned with questions on the conceptual level (and also sanity-checking) our previously established theoretical insights. We've added this to the paper.
> >
> > Regarding the broader impact concerns: we beg to differ here, our paper is not about "taking a causal data set and applying GNN there" but about foundational questions regarding GNN and causality. Naturally, this invokes a philosophical dimension as well.
> >
> > Thank you, we look forward to further discussions. Also, we would like to ask the reviewer to reconsider the usage of terms such as the paper being detrimental to the complete filed of causality. It takes a lot of time to write a paper (as the reviewer will be aware) and using such words in a review do no good for the field or in other words can be detrimental.

---

### Decision · Action_Editors · 2023-08-07

**Recommendation:** Reject

**Comment:**

Reviewer Yjuk supplemented by Reviewer dA5Y sum up the reason for the decision. Both are quoted in full below.

A thoroughly revised version that takes the criticism into account may be considered later.

Reviewer Yjuk:

Overall, this is a very confusing and hard to understand paper due to the consistent conflation of various important concepts pertaining to causality and statistics. Before I go into more detail below, I want to start by complimenting the authors' revisions to the introduction. Now it is much less unstructured and provides a much better onboarding. Although reviewer dA5Y is right to point out that these introductory sections could be shorter overall, I welcome the improved version independent of its length. There are still remaining pockets of unfounded speculation but overall it is a much better read.

One of the most concerning aspects of the paper for me was, and still is, the haphazard use of causal terminology. In my review I pointed out to the authors that their statements re. "non-causal models such as linear regression" does not necessarily make sense, they provided an explanation both in their response and in the text. Their explanation increased my concern as it implied a conflation of two uses of the word "model". The authors basically make the contrast of "non-causal models such as linear regression, CNNs vs. causal models such as SCM". This does not make sense since these refer to two different uses of the word "model": in the former they seem to denote a "function" and in the latter a probabilistic generative model. So the differences between the two do not pertain to the quantification of causal-ness among the two at all. I fear that this kind of conflation is unacceptable for a paper on causality that has theoretical aspirations.

What makes SCM unique as a class of generative model is that it limits randomness to exogenous variables, about which we can conduct posterior inference, and then use the results of this inference to produce alternative scenarios with other (counterfactual) constraints. In that sense it would be understandable comparing SCM's to probabilistic generative models in general, but not to some arbitrary functions, since they are a different class of mathematical objects. The authors do not provide explicit, fulfilling discussions about these concepts and instead use a lot of confusing, vague terms such as "fully" or "partially causal".

Another example of such imprecise language includes "similar to the notions of disentanglement and causality, latent variable models propose the existence of a-priori unknown [variables]..." This is again a comparison or connection between concepts that exist on vastly different abstraction levels. Similar to Reviewer dA5Y's observation on the loose use of the term "dual", the authors justify their statement re. "variational inference overcoming curse of dimensionality" by the newly added explanation "as in making intractable queries tractable". This again is not a helpful use of these terms.

Regarding the authors' main contribution, especially for Section 3, I cannot see the benefit of coaxing a model architecture to emulate an arbitrary probabilistic generative model. We can easily do the same with feedforward neural networks, which can take
 as input and produce
 as output. With some fancy bias nodes, we can reach the same expressivity. On a less important note, the authors also state that "a more radical view would postulate that all GNNs are therefore SCMs": I do not see what is radical nor relevant here. We already established that with arbitrary function definitions we can make GNNs (or another sufficiently flexible modeling formalism) express any generative model SCM's can.

For Section 4, the wider claims of expressivity of GNNs are not pursued, and instead a generative model that can learn from experimental data with interventions is proposed. I think this could be interesting by itself if pursued more deeply, but that would require a much more detailed investigation and comparison with other methods from the literature, on realistic datasets. Also factoring in the concerns of the other reviewers, I unfortunately cannot recommend the acceptance of the paper. I think the use of GNNs is worth exploring in causal inference contexts, and I would encourage the authors' to further their work in this regard, hopefully with the benefit of the reviews provided.

Reviewer dA5Y:

GNNs form a special class of functions. SCMs contain functions in their structural equations. The theoretical results in the paper do not go beyond this well-known fact even after the revision. The experiments are on very small, low-dimensional data that can already be handled easily. Therefore, I was not able to identify a contribution of the paper to make it suitable for publication. The authors disagreed with my request to demonstrate the usefulness of GNNs for causality by solving a causal inference task that cannot be solved otherwise perhaps on high-dimensional graph-structured data, beyond the small examples they used in the experiments. Since I believe the authors did not successfully support their claims on the usefulness of GNNs for causality, I recommend the rejection of the paper.


**Audience:**

Yes.

**Claims And Evidence:**

No. All three reviewers agree that the claims of the paper are not presented in a convincing manner. Reviewer Yjuk sums up the consensus of the reviewers in a clear manner.

**Resubmission Of Major Revision:**

The authors may consider submitting a major revision at a later time.